# Profiling SLAs for cloud system infrastructures and user interactions

M. Emilia Cambronero[1], Adrián Bernal[1], Valentín Valero[1], Pablo C. Cañizares[2] and Alberto Núñez[3]

[1] Albacete Research Institute of Informatics, Department of Computer Science, University of Castilla La Mancha, Albacete, Spain
[2] School of Informatics, Autonomous University of Madrid, Madrid, Spain
[3] School of Informatics, Complutense University of Madrid, Madrid, Spain



## ABSTRACT

Cloud computing has emerged as a cutting-edge technology which is widely used by both private and public institutions, since it eliminates the capital expense of buying, maintaining, and setting up both hardware and software. Clients pay for the services they use, under the so-called Service Level Agreements (SLAs), which are the contracts that establish the terms and costs of the services. In this paper, we propose the CloudCost UML profile, which allows the modeling of cloud architectures and the users' behavior when they interact with the cloud to request resources. We then investigate how to increase the profits of cloud infrastructures by using price schemes. For this purpose, we distinguish between two types of users in the SLAs: *regular* and *high-priority users*. *Regular* users do not require a continuous service, so they can wait to be attended to. In contrast, *high-priority* users require a constant and immediate service, so they pay a greater price for their services. In addition, a computer-aided design tool, called MSCC (Modeling SLAs Cost Cloud), has been implemented to support the CloudCost profile, which enables the creation of specific cloud scenarios, as well as their edition and validation. Finally, we present a complete case study to illustrate the applicability of the CloudCost profile, thus making it possible to draw conclusions about how to increase the profits of the cloud infrastructures studied by adjusting the different cloud parameters and the resource configuration.

## INTRODUCTION

The importance of the cloud has increased enormously over the last few years, and currently it dominates the Information Technology (IT) markets. According to a recent study by the Synergy Research Group (*SRG, 2021*), the cloud market keeps expanding geographically in all regions of the world. Although the public cloud is currently controlled by a few top providers, new opportunities for new participating companies and business models have emerged.

Therefore, it is essential for cloud providers to be more competitive and to have the ability to manage their resources more effectively. With this goal in mind, the modeling and analysis of cloud systems can be a powerful tool for cloud providers to manage their

Corresponding authors
M. Emilia Cambronero,
memilia.cambronero@uclm.es
Adrián Bernal,
adrian.bernal@uclm.es

resources, increase their profits, and be more competitive. Thus, we have defined a UML 2.5 parameterized profile, named CloudCost, to model the architecture and interactions of a cloud system based on Service Level Agreements (SLAs) (*Khan, 2016*). The Unified Modeling Language (UML) (*OMG, 2017*) is one of the most widely recognized and used modeling languages in this field. UML is a standard of the Object Management Group (OMG) (*OMG, 2016*), and is designed to be a general-purpose, developmental, modeling language that provides a standard method to visualize the final design of a system.

Our profile consists of a UML component and a sequence diagram that models the relationships and associations between the system components, the flow of actions, and the interactions between the roles in the system. The main advantage of the use of a parameterized profile is that it makes it possible to specify a wide spectrum of cloud configurations helping cloud service providers to maximize their profit and be more competitive. The main parameters considered are those related to the cost of the different VMs offered and the SLAs signed by the users, such as discounts offered for user subscriptions when their requests cannot be immediately attended to, offers made by the cloud provider to resume execution in the event that it did not finish within the estimated time and compensations due to resource unavailability, in the case of *high-priority* users.

A previous version of the profile was presented in (*Bernal et al., 2019b*). In that work, the cloud infrastructure, the interactions between the users and the cloud provided were modeled without considering any cost-per-use strategy or different user types (SLAs). Thus, the present work is an extension that includes cost-related parameters and two different types of users, namely *regular* users and *high-priority* users, depending on the kind of SLA they sign. *Regular* users do not require a continuous service, so they can wait to be attended to, while *high-priority* users pay for a continuous service, so they need an immediate answer to their requests.

The proposed UML profile captures the main elements of the cloud infrastructure and the client interactions, which is reflected in a methodological way to model different scenarios and then launch the corresponding simulations on a cloud simulator. As testbed, we use the simulator Simcan2Cloud (*Bernal et al., 2019a*), which makes it possible to load the cloud scenarios created by the (Modeling SLAs Cost Cloud) tool and simulate the execution of the workloads generated, which consist of a large number of users. It is worth noting that these simulations can be executed on a personal computer, and therefore no special features are required of the platform to support these executions. The results provided by the simulator allow us to carry out the performance evaluation and the profit analysis of our cloud models.

The paper is structured as follows. The motivation behind, and the main contributions of the paper are explained in "Motivation and Contribution". A complete description of the related work is given in "Related Work", and "Methodology" details the methodology used. "CloudCost Profile" presents the CloudCost UML Profile. "MSCC Design Tool" describes the complete MSCC modeling tool that we have implemented to create, edit, and validate sequence and component models of cloud systems based on SLAs. "Case Study" examines the profile and draws some conclusions about how to increase the cloud profit for the cloud studied by adjusting the different parameters and resource

configuration. Finally, "Conclusions and Future Work" contains the conclusions and future lines of work.

# MOTIVATION AND CONTRIBUTION

In this section, we present the motivation and main contributions of this work.

## Motivation

Most of the existing works about the modeling and analysis of cloud systems are focused on the cloud infrastructure and the performance evaluation from a user's viewpoint. Our goal, however, is to put the focus on the interactions between the users and the cloud service provider in order to analyze the profits obtained by the latter. The results obtained in this work could then be useful to increase these profits by setting the appropriate cloud configuration for an expected workload. Thus, the CloudCost UML profile allows us to model both the cloud infrastructure and the users' interactions with the cloud service provider in order to analyze the profits obtained under different workloads.

For this purpose, the CloudCost UML profile includes the pay-per-use model in cloud systems, by considering two different types of users, namely *regular* and *high-priority* users, who sign the so-called Service Level Agreements (SLAs) to establish the specific conditions and prices to access and use the cloud. *Regular* users request a number of virtual machines from the cloud provider, with some specific characteristics, but they can wait to be attended to when these services are not available. In contrast, *high-priority* users should obtain the services they request immediately, so they usually pay a greater price, and must be compensated when these services cannot be provided.

## Contribution

This paper extends the CloudCost UML profile and the MSCC modeling tool – presented in (*Bernal et al., 2019b*) – by including the new features related to the analysis of profits, namely, the user types, costs per resource, discounts, offers, and compensations. This new version also includes the SLAs, with both types of user, and the users' behavior in terms of their interactions with the cloud service provider for both types of user. Hence, the modeling tool has been extended as well, so that we can easily create and edit parameterized cloud models, so as to consider different cloud infrastructures and different pricing schemes. In addition, these UML models can be validated, and then we can generate the input configuration files required to carry out the performance evaluation using the Simcan2Cloud simulator (*Bernal et al., 2019a*).

To summarize, we can highlight the following main contributions of this paper:

- the definition of a new parameterized UML profile—called CloudCost—for modeling cloud systems with costs, considering the new characteristics of the cost-per-use business model,
- the validation of the parameters assigned in the cloud models using OCL rules,

- the extension of the modeling and validation tool (MSCC), so as to allow us to easily design the cloud infrastructure and the user interactions with SLAs and cost-per-use, and then validate the models,
- a complete case study that illustrates the whole process, from the design of a cloud system to the performance evaluation and profit analysis.

## RELATED WORK

In recent years, there has been a growing interest among the scientific community in cloud services, cloud computing models, SLAs and pricing schemes (*Chi et al., 2017*; *Chen, Lee & Moinzadeh, 2018*; *Soni & Hasan, 2017*; *Cong et al., 2018*). This fact is mainly due to the potential expansion of this computational paradigm and the significance of increasing the cloud service provider's profitability.

*Bokhari, Shallal & Tamandani (2016)* present a comparative study to evaluate the cloud models with the purpose of helping clients to determine what kind of service they need and the risks associated with each model. *Sala-Zárate & Colombo-Mendoza (2012)* present a review of different cloud service providers in the market to help developers, users, and enterprises to select the one which meets their needs. There are other works that focus on a specific cloud model. For instance, *Sharma & Sood (2011)* present an SaaS-oriented work, in which an architecture for defining cloud software services using a Platform-Independent Model (PIM) is introduced. This model is then transformed into one or more Platform-Specific Models (PSMs). The purpose of this paper by Sharma and Sood is to emphasize the benefits of MDA-based software development in developing software applications in the cloud independently of the specific technologies used. An IaaS-oriented work is presented by *Ghosh et al. (2013)*, in which they model a specific class of IaaS cloud to offer services with machines divided into three pools with different values for two parameters: provisioning delay and power consumption. They propose a multi-level interacting stochastic model, in which the model solution is obtained iteratively over individual submodel solutions. From the results obtained in the paper, they state that the workloads and the system characteristics had an impact on two performance measures: mean response delay and job rejection probability.

*Naseri & Jafari Navimipour (2019)* propose a hybrid method for efficient cloud service composition. An agent-based method is also used to compose services by identifying the QoS parameters. Then a particle swarm optimization algorithm is employed to select the best services. They perform several experiments on a simulator implemented in Matlab and analyze the results by considering the number of combined resources, waiting time and the value of a fitness function. The whole process requires a significant time to find a solution. *Zanbouri & Jafari Navimipour (2020)* propose a honeybee mating optimization algorithm for cloud service composition. A trust-based clustering algorithm is used to address the trust challenge. The proposed method is simulated repeatedly with a real workload and a random workload to evaluate its efficiency. It works well for small-scale problems, but its performance with regards to computation time is worse for large-scale problems.

The above works focus on improving the use of cloud services from the user's point of view, while our work aims at increasing the provider's profits without negatively affecting the services offered.

Regarding cloud modeling, different UML profiles have been proposed for modeling multiple aspects of a cloud system. *Kamali, Mohammadi & Barforoush (2014)* present a UML profile to model the deployment of a system in the cloud. It allows the modeling of instances and the infrastructure offered as a service. However, the physical infrastructure, the interactions of the users with the cloud provider and the cost per usage are not considered in that work. *Bergmayr et al. (2014)* propose the Cloud Application Modeling Language (CAML), a UML-based modeling language that considers pricing features, modeling and deployment of cloud topologies. CAML provides dedicated UML profiles in which the cost of the cloud resources is also considered. However, they do not model the physical infrastructure, the user interactions with the cloud provider or the SLAs.

With respect to SLAs, *Papadakis-Vlachopapadopoulos et al. (2019)* propose a collaborative SLA and reputation-based trust management solution for federated cloud environments. It consists of a hybrid reputation system that takes into account both user ratings and monitoring of SLA violations. Some technical KPIs, such as network latency and CPU utilization, are used to measure SLA violations. However, the cost of the services is not considered. *Zhou et al. (2019)* propose a model based on smart contracts to detect and register SLA violations in a trustworthy way using witnesses. They use blockchain to automate the SLA lifecycle and ensure fairness between roles. They deploy the implemented model on a blockchain test net to test all the functionalities. Both provider and customer must reward witnesses for the monitoring service and therefore this entails a cost. This proposal only takes into account the CPU usage and RAM memory, but network features are not considered. *Li et al. (2019)* propose a host overloading/underloading detection algorithm based on a linear regression prediction model to forecast CPU utilization. The goal is to minimize the power consumption and SLA violations by using an SLA-aware and energy-efficient virtual machine consolidation algorithm. They perform several experiments with a real and a random workload on the CloudSim simulator. The authors focus on reducing the energy consumption of the cloud data centers. However, our work tries to increase the income of the cloud service provider through the study of pricing schemes.

Regarding the research works that take into account pricing schemes, we can mention the work by *Chen, Lee & Moinzadeh (2018)*, who conducted a comparative study analyzing two pricing schemes offered to cloud users by some of the biggest cloud service providers: the reservation-based scheme and the utilization-based scheme. The former is also called the R-scheme and is frequently used by Microsoft and Amazon. The latter is also called the U-scheme and is commonly adopted by Google. *Cong et al. (2018)* present a work focused on maximizing the cloud service provider's profits. Their approach analyzes and varies the pricing schemes without violating the established SLA. For this purpose, the authors provide a dynamic pricing model based on the concept of user-perceived value, which captures the real supply and demand relationships in the

**Table 1 Main features of the most relevant approaches.**

| Proposal | SLAs | | Cloud modeling | | | | |
|---|---|---|---|---|---|---|---|
| | Pricing scheme | CP profits | Services | Infrastructure | User interaction | CP | Experiments |
| *Kamali, Mohammadi & Barforoush (2014)* | ✗ | ✗ | ✓ | ✗ | ✗ | ✗ | ✓ |
| *Bergmayr et al. (2014)* | ✓ | ✗ | ✓ | ✓ | ✗ | ✗ | ✗ |
| *Soni & Hasan (2017)* | ✓ | ✗ | ✓ | ✗ | ✗ | ✗ | ✗ |
| *Chen, Lee & Moinzadeh (2018)* | ✓ | ✗ | ✓ | ✗ | ✗ | ✗ | ✓ |
| *Cong et al. (2018)* | ✓ | ✓ | ✓ | ✗ | ✗ | ✓ | ✓ |
| *Herzfeldt et al. (2018)* | ✓ | ✓ | ✓ | ✗ | ✗ | ✓ | ✗ |
| CloudCost & MSCC (2021) | ✓ | ✓ | ✓ | ✓ | ✓ | ✓ | ✓ |

cloud service market. *Soni & Hasan (2017)* present a pricing scheme comparison based on several characteristics, such as fairness, merits, and demerits. In this study, the authors include a discussion related to both service and deployment models. All of these studies analyze several pricing schemes offered to the users, with the main objective of maximizing the cloud service provider profit, while offering better prices to the users for some services. However, the main goal of our work, beyond the study of pricing schemes, is to analyze and increase the profitability of the cloud service provider while maintaining a balance between the cost of the infrastructure and the user's demands.

*Herzfeldt et al. (2018)* discuss different guidelines for the profitable design of cloud services. They carried out 14 interviews with cloud service provider experts, in which they addressed the relationship between value facilitation, that is, the capability to accumulate resources for future customer demands, and profitability for the cloud service provider. In the present work, we adopt the second perspective, that is, the study of the profitability for the cloud service provider. However, we should point out that our approach is quite different from the above works. We model both the cloud infrastructure and the user interactions with the cloud service provider, with the goal of analyzing how they affect the global incomes for the cloud service provider.

For the purpose of showing the main differences between this paper and the existing works, as well as presenting the main novelties of our proposal, we have conducted a comparison between some of the most relevant approaches analyzed in this section and our work (see Table 1). The first column of the table shows the authors of the proposal. The next two columns, namely *Pricing scheme* and *CP profits*, concern aspects related to SLAs. Specifically, the former indicates whether the proposal analyzed provides some type of pricing scheme. As can be seen, all the papers reviewed except *Kamali, Mohammadi & Barforoush (2014)* provide it. The latter shows whether the approaches are aimed at enhancing the profits of the cloud service provider. In this case, *Soni & Hasan (2017)*, *Cong et al. (2018)*, *Herzfeldt et al. (2018)*, and our work provide this feature. The following five columns focus on *Cloud modeling* aspects. *Services* indicates whether the proposal is able to model cloud services. This feature is actually supported by all the works in the table. *Infrastructure* denotes whether the work provides the mechanisms to

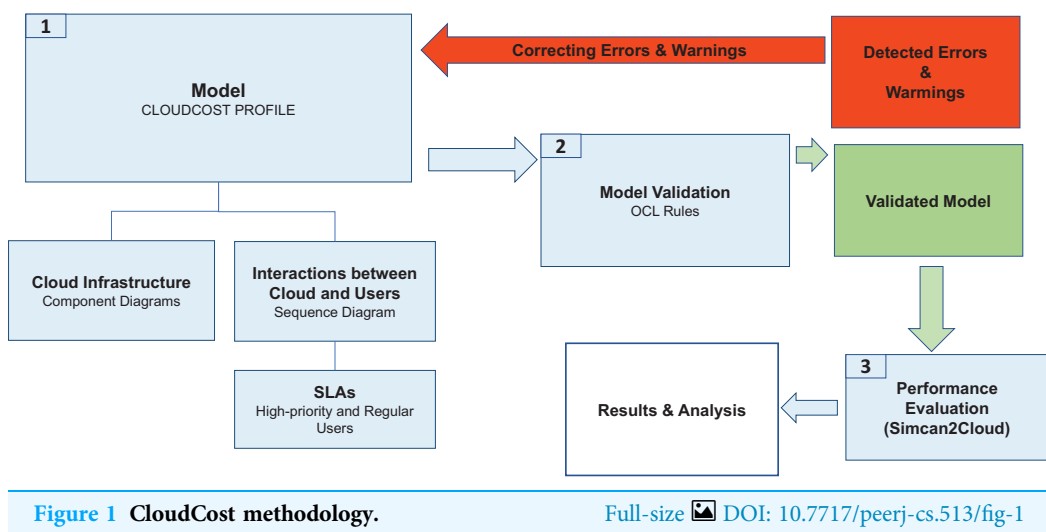

**Figure 1  CloudCost methodology.**

model a cloud computing infrastructure. In this case, besides the current work, only *Kamali, Mohammadi & Barforoush (2014)* through UML profiles and *Bergmayr et al. (2014)* via the CAML language support modeling infrastructures. *User interaction* indicates whether the proposal supports modeling the communications between the users and the cloud service provider. This feature is only supported by our work. Next, the column labeled *CP* shows whether the cloud service provider's behavior is modeled with a high level of detail. In this case, only *Cong et al. (2018)*, *Herzfeldt et al. (2018)* and our proposal include this feature. Finally, the last column denotes whether experiments were conducted in the study. With regard to experimental studies, *Kamali, Mohammadi & Barforoush (2014)* deploy a bank management's system in a cloud computing environment based on their profile. *Chen, Lee & Moinzadeh (2018)* conduct a numerical study to examine the impact of the pricing model parameters on the service providers' optimal pricing decisions and social welfare. *Cong et al. (2018)* conduct simulation experiments with Matlab to test the effectiveness of the proposed scheme based on the user-perceived value. Finally, in our work, we perform simulations with Simcan2Cloud to draw conclusions about how to improve the management of resources to increase the provider's profits, without negatively affecting the services offered.

# METHODOLOGY

In this section, we describe the methodology used to model and analyze the performance of cloud system infrastructures by considering SLAs for two different types of user (*regular* and *high-priority*). Figure 1 shows the different phases of this methodology, which are described as follows:

1. System Modeling. The CloudCost UML profile is defined to model both the cloud infrastructure and the interactions between the cloud service provider and the users when they access a cloud to request resources. This profile consists of sequence and component diagrams. As the behavior of *regular* and *high-priority* users is different, we consider two parameterized sequence diagrams, one for each type of user. These

diagrams show the interactions of the users with the cloud service provider, thus defining the behavior of each type of user. In addition, the number of virtual machines required by the users and their specifications are established by setting up the corresponding parameters in these diagrams. Furthermore, a component diagram is used to model the infrastructure of the cloud system. The specific cloud infrastructure configuration is then established by setting the corresponding parameters in the component diagram.

2. Model Validation. Each cloud model generated from the profile is validated to check certain properties they must fulfill. For instance, the costs must be greater than or equal to 0, the users must sign one SLA, etc. A set of OCL rules are defined for this purpose, and as a result of this validation we obtain the possible errors or warnings in the model. If there are errors in the model, they must be fixed, so we return again to Phase 1 to correct the model, and then we must validate it again (Phase 2).

3. Performance Evaluation. Once the model has been validated, the configuration files are generated for the cloud simulator (Simcan2Cloud (*Bernal et al., 2019a*)). Simulations are then executed, providing us with the performance metrics, namely the number of *regular/high-priority* users that were served, the number of them that left the system without being served, the waiting times for the users, etc.

The analysis of these results allows us to draw relevant conclusions about the most appropriate cloud infrastructure for a specific workload.

# CLOUDCOST PROFILE

In this section we define the CloudCost profile, which is an extension of the Model4Cloud profile that we introduced in (*Bernal et al., 2019b*), including costs and SLAs for two types of user (*regular* and *high-priority*).

## CloudCost profile

Users are classified into two types, namely *regular* and *high-priority*, and they request certain VM resources, according to the catalog offered by the cloud provider. *Regular* users do not require an immediate answer to their requests, so they can wait to be attended to, and thus the price they pay varies depending not only on the VM features they have requested, but also on the conditions in which they are finally provided with them. In contrast, *high-priority* users expect an immediate answer to their requests, in some cases on a 24/7 basis, so they are able to pay for extra resources (if required) for their services to be immediately executed. It should be very unlikely for a *high-priority* user request not to be met, and a compensation must be offered in this case.

We consider that a cloud infrastructure consists of one or several data centers, each of which consists of a set of nodes grouped in racks that are interconnected through a communication network. Each rack contains a collection of nodes with the same hardware features, that is, CPU, memory, and storage. All this infrastructure is managed by a cloud service provider that offers a catalog of VMs with assigned Service Level Agreements

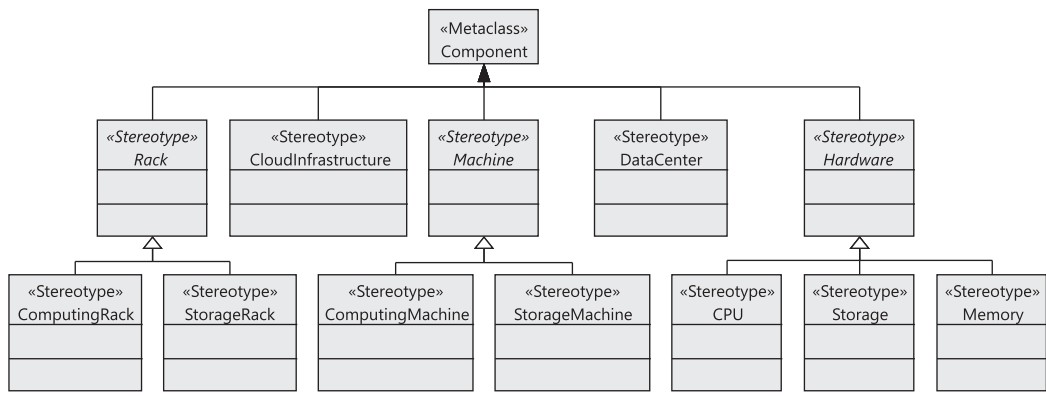

**Figure 2** CloudCost profile: cloud infrastructure stereotypes.

(SLAs), which include the service costs to rent these resources. These VMs are mapped to physical machines by using a specific resource allocation policy.

Some of these machines are always available, ready to serve a user request, but there are some reserved physical machines that will only be used for *high-priority* user requests. Thus, the cloud service provider reserves some machines to be only used when *high-priority* user requests cannot be met by the set of *normal* (non-reserved and always in execution) machines. In such a case, one of the reserved machines with enough resources to fulfill the user demands must be activated to satisfy the request. It is a critical decision for the cloud service provider to fix the number and features of the available machines to attend to the *regular* user requests, as well as to define the ratio of reserved machines that will attend to the incoming requests from *high-priority* users when they cannot be served by the non-reserved machines. In this decision, the cloud service provider must take into account both the total number of available physical resources and the workload generated by the users, in order to attend to the largest number of users.

### Component and sequence diagram

Figure 2 shows the stereotypes defined to model the components of the cloud infrastructure, that is, data centers, storage and computing racks, number of machines in the racks, CPUs, and memories. The *CloudInfrastructure* stereotype extends the *Component* metaclass and represents the infrastructure managed by the cloud service provider. The *Rack*, *Machine*, *DataCenter*, and *Hardware* stereotypes also extend the *Component* metaclass. The cloud infrastructure consists of a collection of data centers. Each data center is equipped with a collection of racks, which in turn consist of a collection of computing or storage machines, which are represented by the *ComputingRack* and *StorageRack* stereotypes, respectively. A computing rack consists of a set of computing machines (*ComputingMachine*), and finally a storage rack consists of a set of storage machines (*StorageMachine*).

Many components of a data center normally have the same characteristics, as they are usually purchased in large quantities. Therefore, we have defined the relationships between components as associations between stereotypes (see Figure 3), so that each

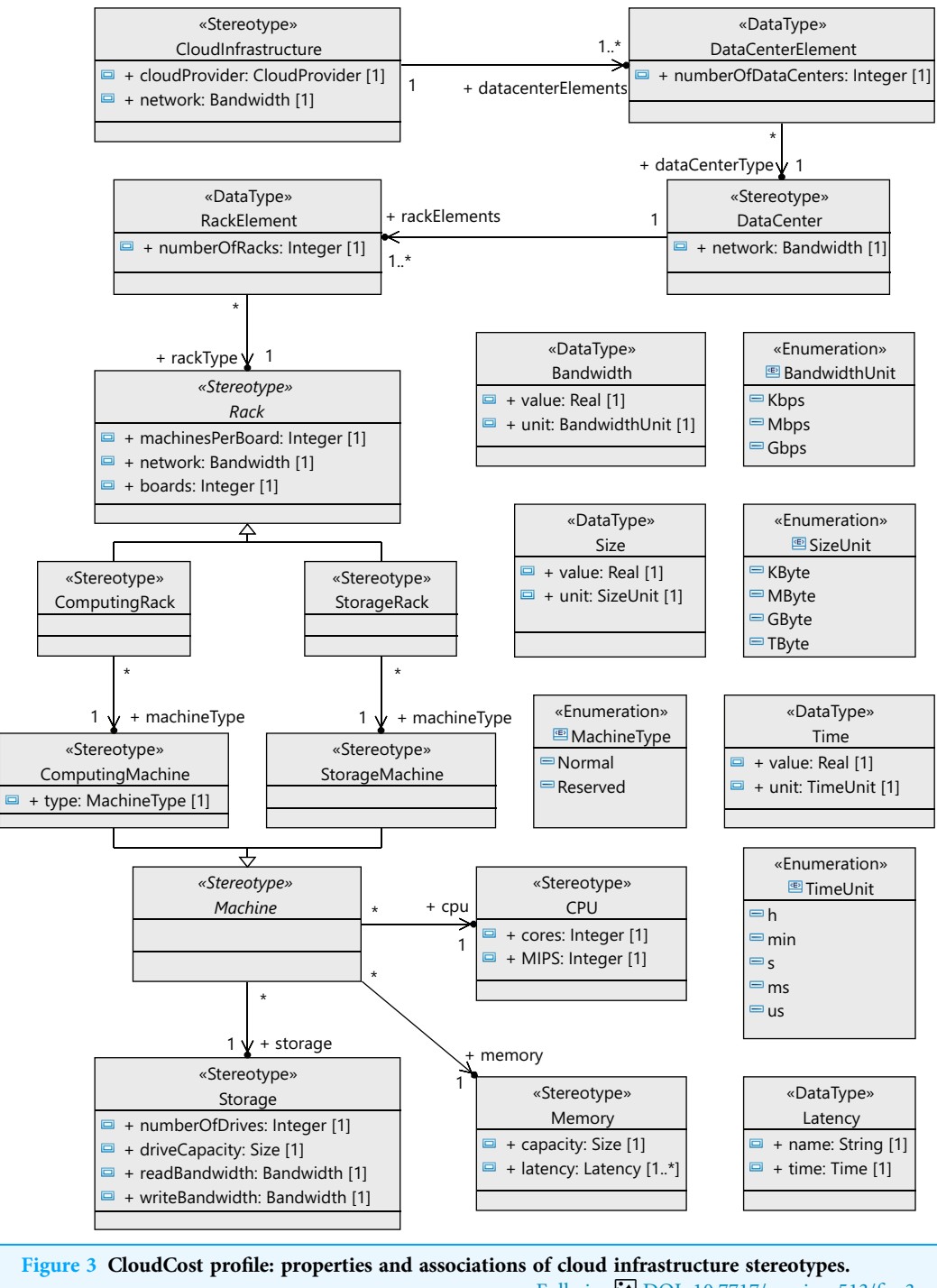

**Figure 3 CloudCost profile: properties and associations of cloud infrastructure stereotypes.**

component can be referenced from different places and reused. We can see in Figure 3 that a *CloudInfrastructure* consists of a cloud service provider, which manages a number of data centers. The *DataCenterElement* data type represents a collection of data centers with the same configuration.

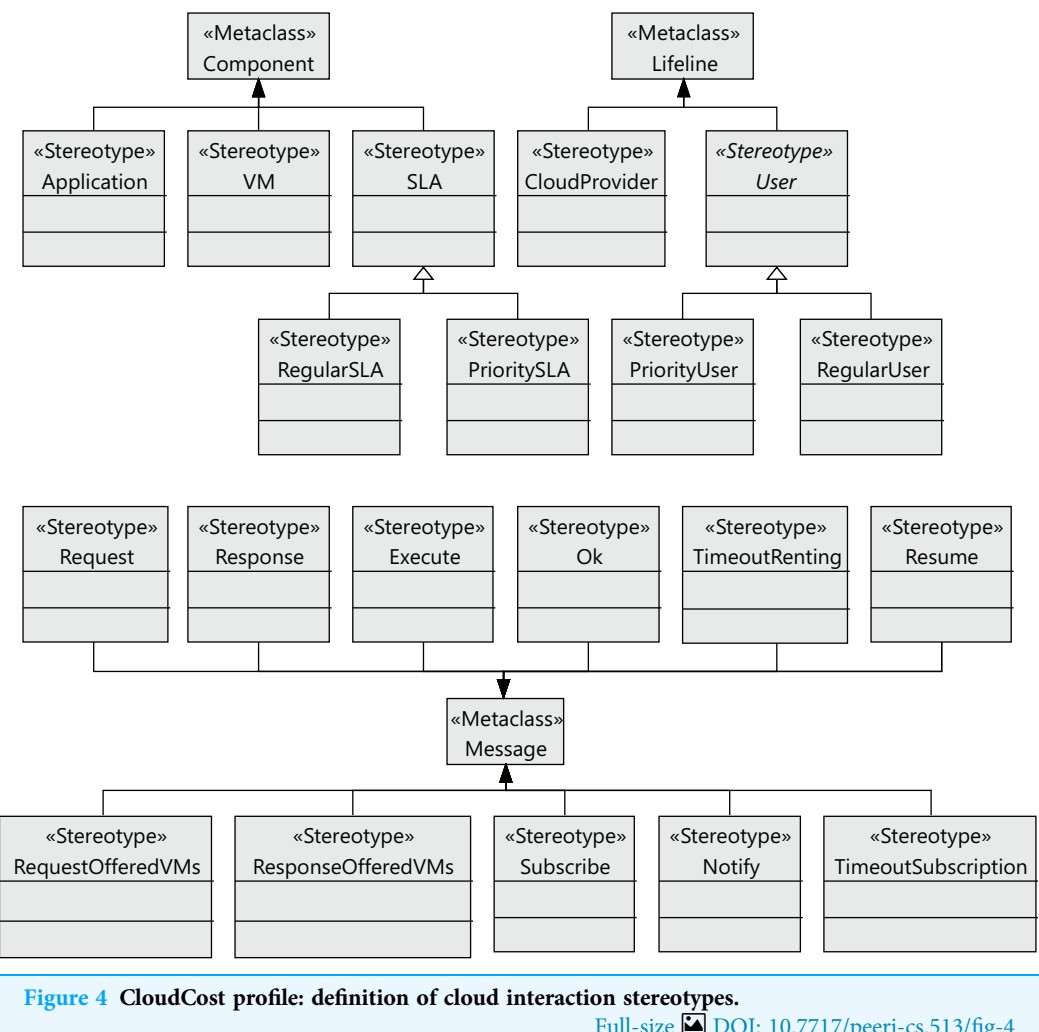

**Figure 4  CloudCost profile: definition of cloud interaction stereotypes.**

In the same way, the *RackElement* data type has been modeled to represent a collection of racks with the same configuration. As an illustration, in the rack specification, we must define the number of boards, the number of machines per board and the network bandwidth for the communication between machines. Furthermore, computing machines can be either *non-reserved* or *reserved*, as indicated above.

The following step to define the profile is to define the stereotypes for the interactions between the users and the cloud provider (see Figure 4). Users request virtual machines (*VM* stereotype) with their associated SLAs (*SLA* stereotype), and the execution of applications on the virtual machines (*Application* stereotype) extends the *Component* metaclass. Both the users and the cloud service provider have behaviors that follow a lifeline (*Lifeline* metaclass). All the messages exchanged extend the *Message* metaclass. The relationships between these components are shown in Figure 5 as stereotype associations. As an illustration, we can see the different SLAs offered by the cloud service provider with the cost of the VMs and signed by the users (*User* stereotype). A VM request

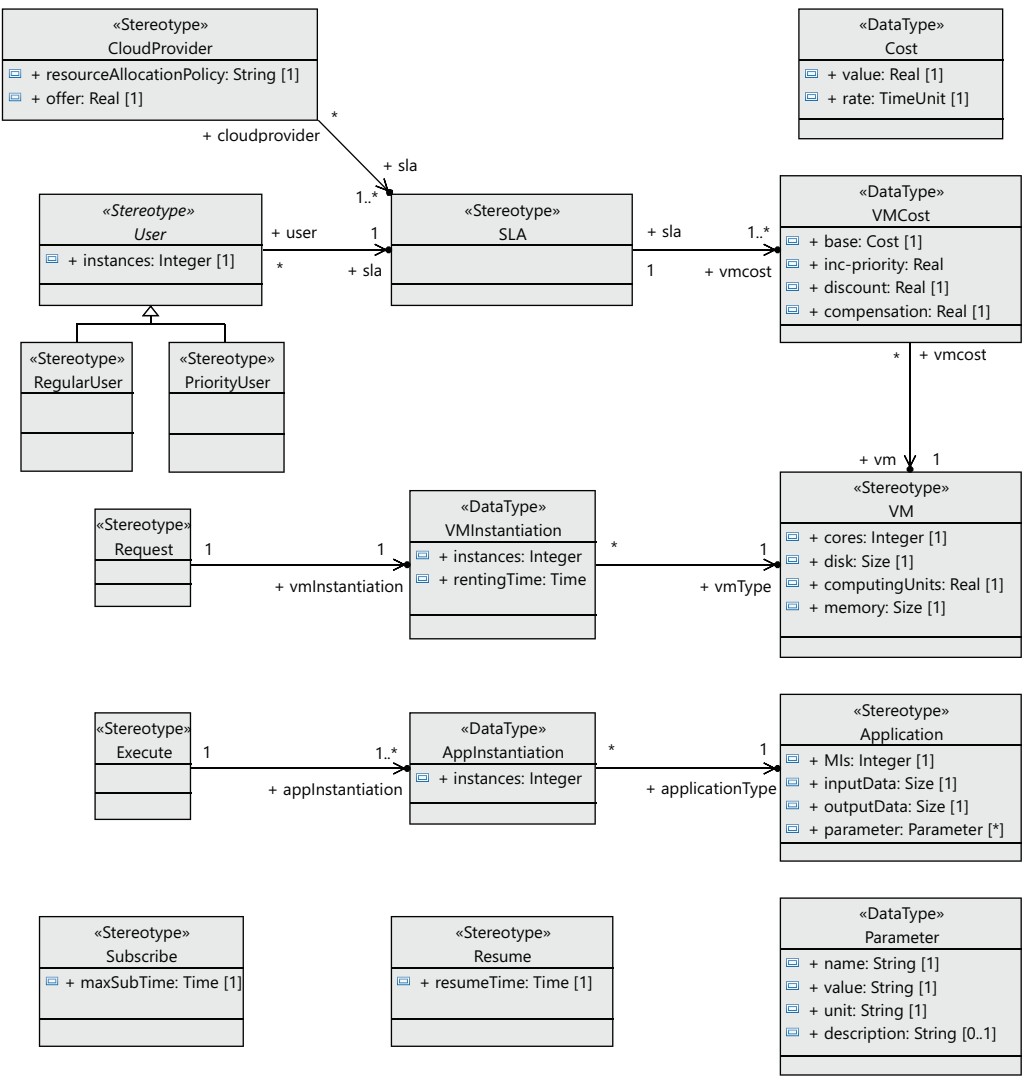

**Figure 5 CloudCost profile: associations and properties of cloud interaction stereotypes.**

consists of the following elements: number of cores, computing units (CUs) for the CPU cores, disk size and memory size.

A.- UML profile for *regular* users.

Figure 6 shows the sequence diagram for the *regular* user's behavior. In this diagram, we capture the interactions of a *regular* user with the cloud service provider when renting a VM. First, the user requests the list of VMs offered (*requestOfferedVms* message) by the cloud service provider in order to know which of them fits his needs best. In response, the cloud service provider sends the list of available VMs to the user. In this message, the cloud service provider indicates the attributes of each VM: CPUs, storage, memory and base cost per hour defined in the SLA. This base cost is the amount to be paid for one hour of the VM under normal conditions when a request can be immediately attended to.

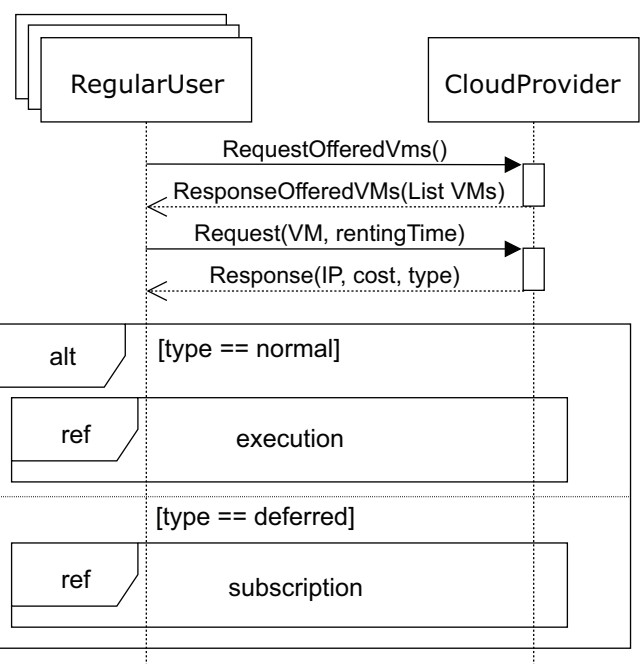

**Figure 6 Main SD: cloud provider and *regular* users interaction.**

However, if no VM is currently available to provide the service (with the user's required features), the user will be informed and receive a discount for the delay.

Thus, with the list of VMs the user requests (*request* message) one of these VMs for a time (*rentingTime*). Afterwards, the cloud service provider responds with the IP, type, and cost of the allocated rental VM. When the request can be immediately satisfied, the type will be *normal*, otherwise, the type returned will be *deferred*. In both cases, the corresponding cost is also returned. Obviously, the deferred price will be lower than the normal price, because the user has to wait for the service to become available.

After receiving a *normal* type answer, the user can execute the applications on the VM provided (see Fig. 7). Two cases can now arise: either the execution of the applications finishes on time, so the user receives an *ok* message from the cloud service provider and the interaction terminates, or the renting time expires before the applications have been completely executed. In this case, the cloud provider offers the user an extension to the renting time with the base price per hour plus a surcharge (*offer*), i.e. the user can pay for this extra time in order to complete the execution, or the user can decline and stop the interactions (see Fig. 8).

Finally, when no VM with the required features becomes available, the user receives the *deferred* message (see Fig. 6). As mentioned above, the price, in this case, will be lower, so the VM renting price will have a discount applied to the normal cost. The user can now decide to wait for the required VM to become available or leave. If the user decides to wait, he subscribes to the VM characteristics (see Fig. 9) for a specific time (*maxSubTime*), with the intention of being notified when a VM with these features is available, and then, upon receiving this notification message, the user starts the execution

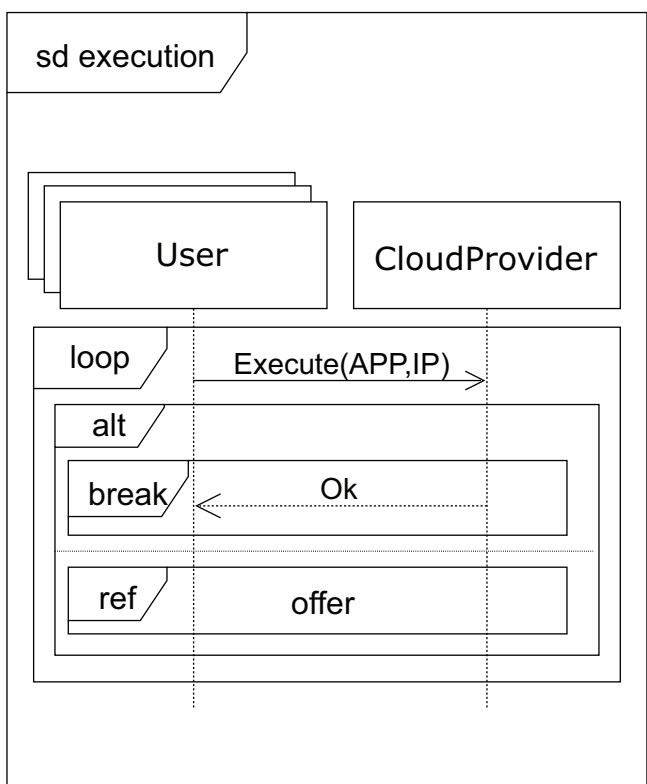

**Figure 7 Execution SD: users and cloud service provider interactions to submit the applications.**

of his applications. Note that when the subscription time expires and no VM has been available for this period, the user leaves without being able to execute the applications.

B.- UML profile for *high-priority* users.

The sequence diagram for *high-priority* users is shown in Fig. 10. The SLA for these users states that they should obtain the requested services immediately, and if no VM matching their needs is available at that moment, the cloud service provider must start up a VM in order to provide the service. In the unlikely event that the cloud service provider cannot start up a VM with the requested features, the user must be compensated for the damages caused. This case would only occur when a VM with the requested features in the pool of reserved machines cannot be allocated, which would be caused by an unexpected number of *high-priority* user requests. This would be a consequence of a misconfiguration of the cloud, and would probably require the addition of new racks in order to be able to deploy some additional VMs while keeping the system well balanced.

Like the *regular* users, *high-priority* users request (message *requestOfferedVms*) the list of VMs from the cloud service provider. The cloud service provider replies with the list of VMs (message *response*), indicating the corresponding costs per hour for each one of them. The user then requests one for a certain period of time (argument *rentingTime* in message *request*). If the requested VM is available, the user executes (see Fig. 7) his applications, paying the amount indicated. Otherwise, if there is no available VM with the requested features, the cloud service provider should start up one of the reserved machines

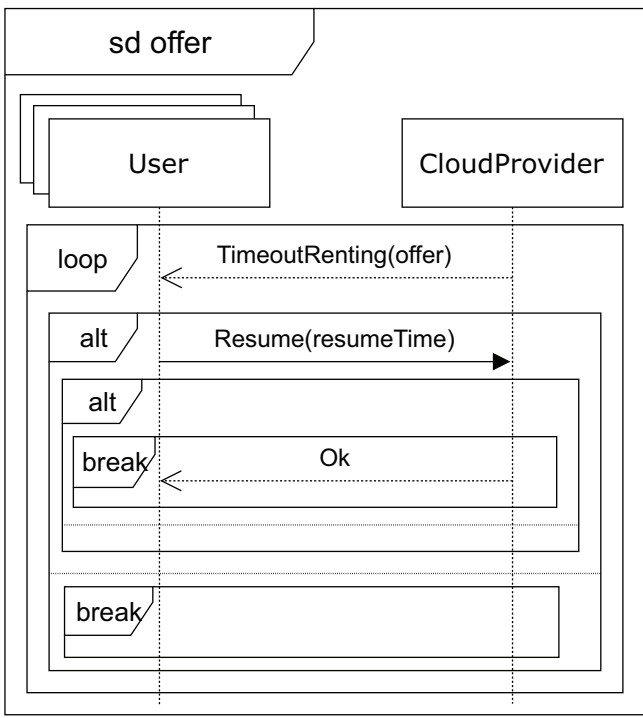

**Figure 8 Offer SD: users and cloud service provider interactions when a time-out occurs.**

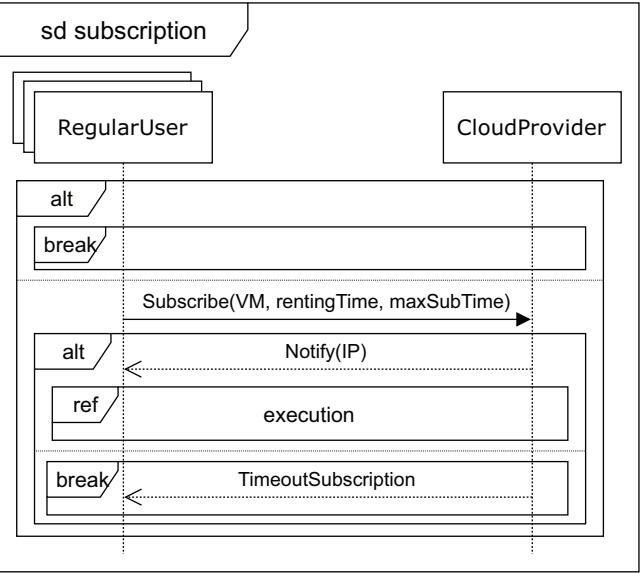

**Figure 9 Subscription SD:** *regular* **users and cloud service provider interactions for deferred execution.**

with the VM matching the user's requirements. In this case, the user must pay a surcharge, which is included in the cost indicated in the *response* message. As mentioned above, should the cloud service provider not be able to start up a VM matching the user's

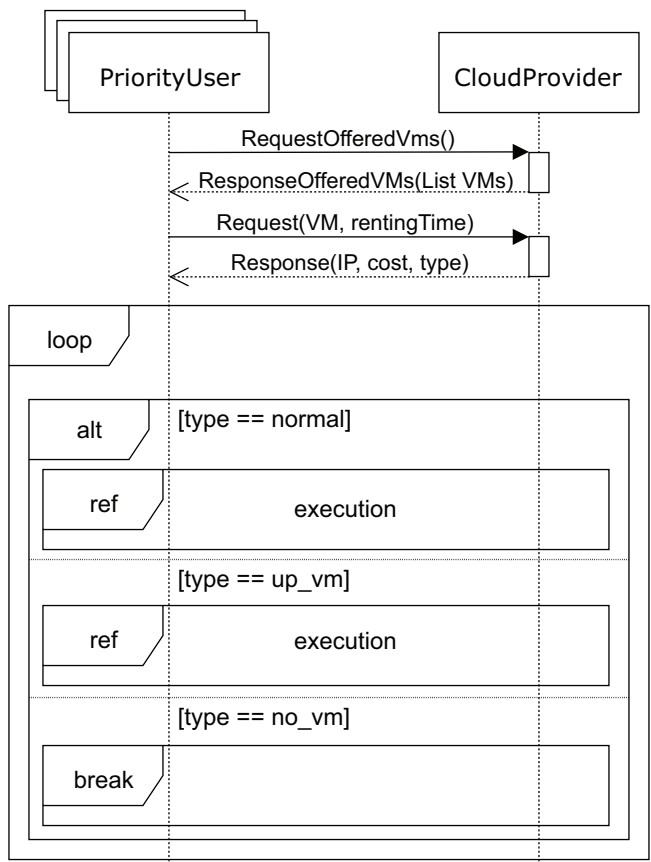

**Figure 10 Main SD: cloud provider and *high-priority* users interaction.**

requirements, compensation must be provided. In this case, the interactions stop (*no_vm* in Fig. 10), and the user receives an economic compensation, as indicated in the SLA.

## Validation of CloudCost profile models

The relationships between stereotypes and their properties define constraints by themselves (see Fig. 5). For instance, a cloud service provider must offer at least one SLA, where the SLA must at least have the cost of a VM, but this SLA could include more than one VM. However, some constraints cannot be defined and checked through stereotype relationships. Thus, they have to be explicitly checked to ensure the model's correctness. Our proposed MSCC (Modeling SLAs Cost Cloud) tool also makes it possible to validate the model by defining a set of Object Constraint Language (OCL) rules (*OMG, 2014*). OCL is a declarative language designed to specify detailed aspects of a system designed using UML models, and it is now part of the UML standard. OCL is considered a formal specification constraint language for UML, which allows us to define object query expressions in our UML models, and to carry out the validation of the CloudCost pro file. Tables 2, 3 and 4 present the OCL rules that have been considered in order to check the model's correctness.

**Table 2 OCL rules for the model validation process I.**

| | |
|---|---|
| **Rule 1** | **CloudProvider_must_offer_once_each_VM_for_each_SLA_type** |
| Rule Description | Cloud service provider must offer each VM only once for each SLA type. |
| Rule OCL Code | context SLA<br>  inv self.vmcost->isUnique(vmcost \| vmcost.vm) |
| Notification Type | Error |
| MSCC Recommendation | Please, enter each VM only once in each SLA! |
| **Rule 2** | **User_must_sign_a_VM_SLA_offered_by_the_CloudProvider** |
| Rule Description | A User must always sign an SLA offered by the cloud service provider. |
| Rule OCL Code | context User<br>  inv uml2cloud::CloudProvider.allInstances()<br>  ->collect(cp \| cp.sla)->includes(self.sla) |
| Notification Type | Error |
| MSCC Recommendation | Please, sign an SLA! |
| **Rule 3** | **User_request_a_VM_not_offered_in_the_signed_SLA** |
| Rule Description | A user cannot request a VM, which is not offered in the signed SLA. |
| Rule OCL Code | context Request<br>  inv uml2cloud::User.allInstances()<br>  ->select(user \| user.base_Lifeline.coveredBy<br>  ->includes(self.sendEvent))<br>  ->collect(user \| user.sla.vmcost<br>  ->collect(vmcost \| vmcost.vm))<br>  ->includes(self.vmInstantation.vmType) |
| Notification Type | Error |
| MSCC Recommendation | Please, one of the following actions must be performed to resolve the error: 1. A user must request another VM that is in the SLA. 2. A user must sign another SLA including that VM type. 3. The cloud service provider must include that type in this SLA signed by the user! |
| **Rule 4** | **SLA_base_cost_must_be_greater_than_or_equal_to_zero** |
| Rule Description | The defined base cost value must be greater or equal to zero. |
| Rule OCL Code | context SLA<br>  inv self.vmcost<br>  ->forAll(vmcost \| vmcost.base.value >= 0) |
| Notification Type | Error |
| MSCC Recommendation | Please, enter a positive value for the base cost value! |
| **Rule 5** | **SLA_inc-priority_cost_must_be_greater_than_or_equal_to_zero** |
| Rule Description | The defined inc-priority cost for *high-priority* users must be greater or equal to zero. |
| Rule OCL Code | context SLA<br>  inv self.vmcost<br>  ->forAll(vmcost \| vmcost.incpriority >= 0) |
| Notification Type | Error |
| MSCC Recommendation | Please, enter a positive value for the *inc-cost* cost value! |

(Continued)

| Rule 6 | SLA_discount_must_be_greater_than_or_equal_to_zero |
|---|---|
| Rule Description | The discount offered to *regular* users must be greater or equal to zero. |
| Rule OCL Code | context SLA<br>  inv self.vmcost<br>   ->forAll(vmcost \| vmcost.discount >= 0) |
| Notification Type | Error |
| MSCC Recommendation | Please, enter a positive value for the *discount* value! |

**Table 3** OCL rules for the model validation process II.

| Rule 7 | SLA_compensation_must_be_greater_than_or_equal_to_zero |
|---|---|
| Rule Description | The compensation cost for *high-priority* users must be greater or equal to zero. |
| Rule OCL Code | context SLA<br>  inv self.vmcost<br>   ->forAll(vmcost \| vmcost.compensation >= 0) |
| Notification Type | Error |
| MSCC Recommendation | Please, enter a positive value for the *compensation* cost value! |
| **Rule 8** | **CloudProvider_offer_must_be_greater_than_or_equal_to_zero** |
| Rule OCL Code | context CloudProvider<br>  inv self.offer >= 0 |
| Notification Type | Error |
| MSCC Recommendation | Please, enter a positive value for the *offer* cost value! |

Tables 2 and 3 show the OCL rules for detecting errors in the parameterization process. The first rule checks that the cloud service provider does not offer a specific VM more than once in the same SLA, since this could give rise to a situation in which a VM would have two different cost values for the same user. The second rule ensures that a user always signs an SLA offered by the cloud service provider. The third rule validates that the user requests a VM that is actually in the signed SLA. Rules 4 to 8 check that the costs defined in the SLA have positive values. OCL also makes it possible define restrictions in the model's behavior to show recommendations to the users in order to parameterize the model. These recommendations appear as *warnings* in the validation process. Specifically, we have defined two possible *warnings*, which could be launched during the validation process (see rules 9 and 10 in Table 4). Rule 9 launches a *warning* when there are *high-priority* users making requests, but there are no machines reserved for them. Rule 10 checks whether the resume time for a VM is set too long compared with the renting time, since we consider that the user will probably want to rent the VM for longer than in the first request.

As an example of validation, let us consider the situation presented in Fig. 11, which shows a fragment of an interaction diagram between the user and the cloud service

**Table 4 OCL rules for the model validation process III.**

| Rule 9 | PriorityUser_has_been_modeled_but_no_machine_has_been_reserved |
|---|---|
| Rule Description | There are no reserved VMs for the *high-priority* user. |
| Rule OCL Code | context PriorityUser<br>  inv uml2cloud::CloudInfrastructure.allInstances()<br>  ->collect(ci \| ci.datacenterElements)<br>  ->collect(de \| de.dataCenterType)<br>  ->collect(dc \| dc.rackElements)<br>  ->select(re \| re.rackType .oclIsKindOf(uml2cloud::ComputingRack)<br>  ->select(re \| re.rackType .oclAsType(uml2cloud::ComputingRack) .machineType.<br>  type= uml2cloud::MachineType::Reserved)<br>  ->collectNested(re \| re.numberOfRacks*re.rackType.boards *re.rackType.<br>  machinesPerBoard)->sum()>0 |
| Notification Type | Warning |
| MSCC Recommendation | Please, reserve some VMs for the *high-priority* user! |
| **Rule 10** | **ResumeTime_is_at_least_twice_as_long_as_the_renting_time** |
| Rule Description | The initial request time for a VM should be longer than the Renting Time. |
| Rule OCL Code | context Resume<br>  inv uml2cloud::Request.allInstances()<br>  ->select(req \| uml2cloud::RegularUser.allInstances()<br>  ->select(ru \| ru.base_Lifeline.coveredBy<br>  ->includes(self.base_Message.sendEvent))<br>  ->forAll( ru \| ru.base_Lifeline.coveredBy<br>  ->includes(req.base_Message.sendEvent)))<br>  ->forAll(req \| req.vmInstantiation.rentingTime .value*2>self.resumeTime.value) |
| Notification Type | Warning |
| MSCC Recommendation | Please, consider that the initial request time should be longer than the renting time! |

provider. The user requests a VM of the type *VM_xlarge*. If the cloud service provider does not offer this VM type in the SLA signed by the user, the third OCL rule in Table 2 will be violated. As a consequence, an error message (Constraint *User_Must_Sign_A_VM_SLA_Offered_By_The_CloudProvider is violated*) will be displayed by the MSCC tool, as can be seen in the figure. The user has several options to solve the error: the user can request another VM that is in the SLA, or sign another SLA including the *VM_xlarge* type, or the cloud service provider could include the *VM_xlarge* type in the existing SLA. In Fig. 11, we can see that the MSCC tool launches a warning when the user sets the resume time to 10 hours. The renting time in the initial request was 2 hours, but the user applications did not finish their execution in that time, so the user decides to resume for a further 10 hours. As a consequence, the last rule in Table 4 is violated, and the MSCC tool shows a *warning* message. In order to address this *warning*, the user should initially rent the VM for a longer period (*renting Time*) and reduce the resumption time (*resume Time*).

Figure 12 shows the validation of a component diagram. In this case, the user has established a negative value for the base cost of a VM of *VM_nano* type. As a consequence,

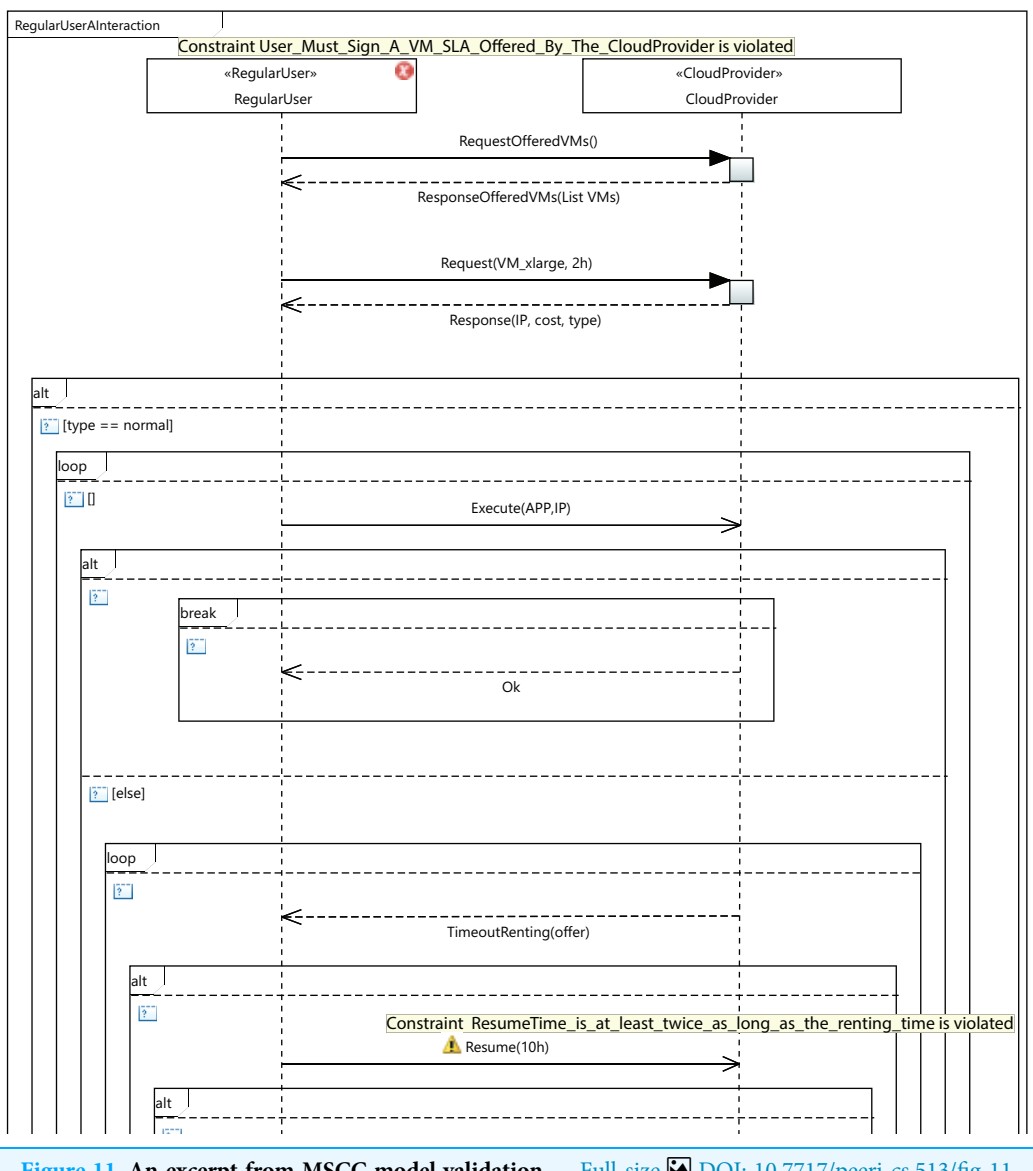

**Figure 11 An excerpt from MSCC model validation.**

the fourth rule in Table 2 launches a violation, and the MSCC tool shows an error. Then, to address this *warning* the user should set a positive value for all the VM costs.

## MSCC DESIGN TOOL

This section presents the MSCC (Modeling SLAs Cost Cloud) computer-aided design tool[1]. This tool focuses on the modeling of cloud systems, considering SLAs to define different user types, and the resources that can be provided for a given cost. Therefore, as mentioned above, this tool allows the user to parameterize the CloudCost profile to establish the value of certain parameters, such as the VMs requested by the user, the waiting time for *regular* users, and the compensation for *high-priority* users, and then to validate the models.

[1] MSCC is available at: https://www.dsi.uclm.es/cloud/modeling/uml2cloud/releases/2.1

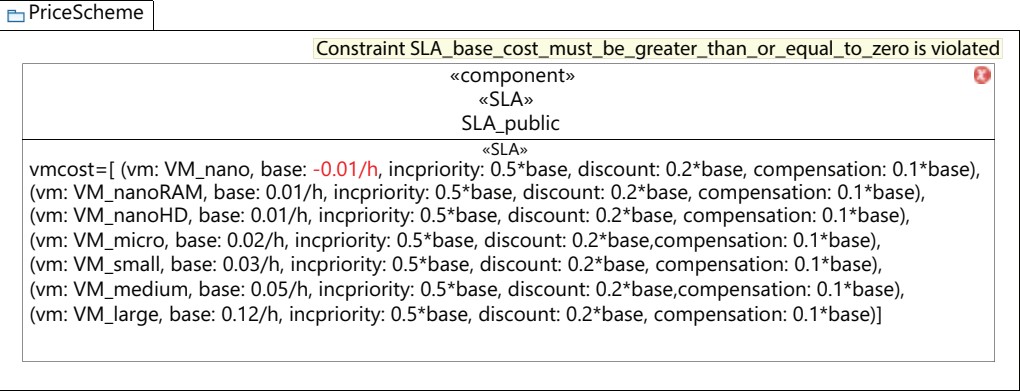

**Figure 12** **MSCC Model validation results.**

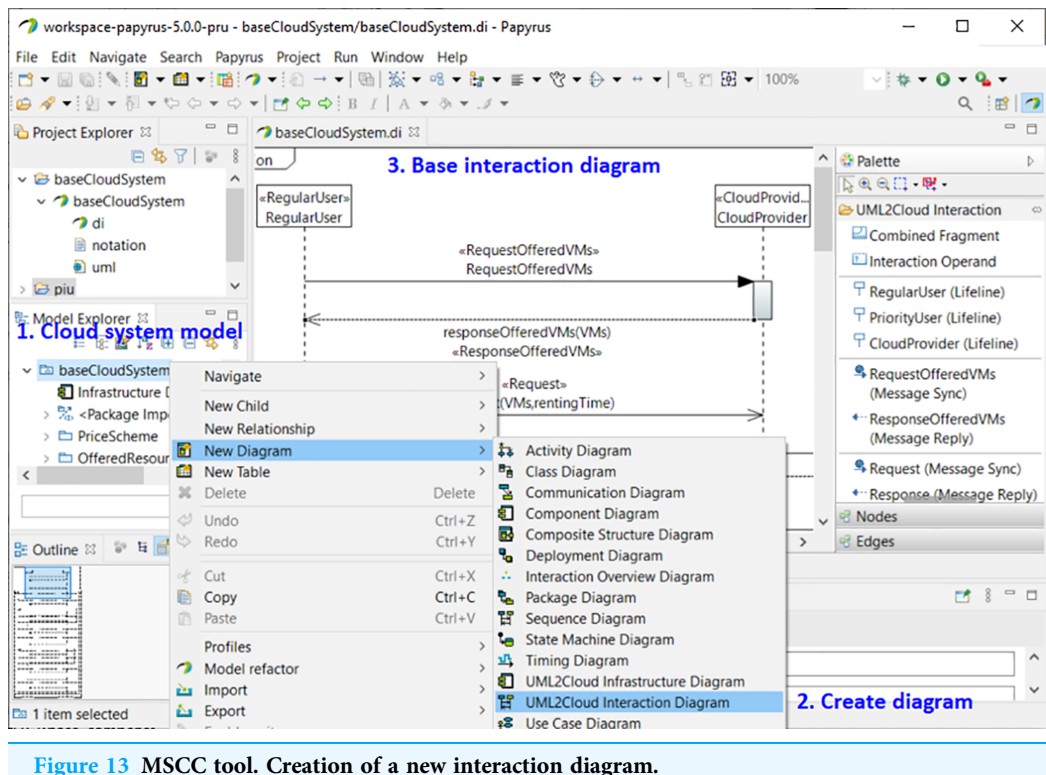

**Figure 13** **MSCC tool. Creation of a new interaction diagram.**

Figure 13 shows a screenshot from the MSCC tool, which is an extension of the tool presented in a previous work (*Bernal et al., 2019b*), and has been adapted to the new UML profile, as described in "CloudCost Profile". It has been implemented using Papyrus (*Gérard et al., 2010*), which is an open-source Eclipse-based tool that provides an integrated environment for creating and editing UML models. The plug-ins used in the

previous version have been adapted in order to include the new functionalities. These new features are the following:

- **es.uclm.uml2cloud.profile**: this plug-in includes the new UML profile, which takes into account both types of user, namely *regular* and *high-priority*, and the resource costs, in order to parameterize different cloud configurations and user interactions.
- **es.uclm.uml2cloud.validation**: this plug-in implements the constraints mentioned in "Validation of CloudCost Profile Models" in order to validate the model.
- **es.uclm.uml2cloud.customization**: with this plug-in, the property views and the tool palettes have been extended to suit the new stereotypes.
- **es.uclm.uml2cloud.examples**: this plug-in contains the examples that have been used to illustrate the applicability of the MSCC modeling tool.

Figure 13 shows how to create a new interaction diagram with the MSCC tool. For the sake of clarity, we have included annotations in the figure. Annotation 1 shows how to select one of the pre-installed examples of the tool. In Annotation 2 we can see the selection of the interaction diagram. Afterwards, another window allows us to select the type of user. Finally, the end-user only has to set the parameters of the automatically-created base interaction diagram corresponding to the selected type of user (Annotation 3).

## CASE STUDY

This section provides a case study that shows the applicability of both CloudCost, our proposed UML profile for representing the users' behavior in cloud environments, and MSCC, a tool for modeling cloud infrastructures. In essence, we are interested in analyzing the overall cloud income for processing the requests of a large number of users (workload) when different data-centers—supporting the cloud—are used. The workloads are generated using, as their basis, two different user roles: *regular* users and *high-priority* users. The experiments in this study were run on the Simcan2Cloud simulator (*Bernal et al., 2019a*). In summary, the process for carrying out the experiments consists of the following steps: (1) modeling five cloud environments using MSCC; (2) generating the configuration files representing these clouds for the Simcan2Cloud simulator; (3) encoding the behavior of the users represented in "CloudCost Profile" into Simcan2Cloud; and (4) simulating the processing of each workload in the five cloud environments modeled.

In order to clearly present this case study, the rest of this section is structured as follows. Firstly, we describe—in "Experimental Settings"—how each part of the cloud environment, that is, the underlying cloud architecture and the workloads, were modeled. Next, in "Performance Analysis", we analyze these models by simulating different cloud scenarios. Finally, we draw conclusions from the results obtained in "Discussion of the Results".

### Experimental settings

In order to conduct the experimental study, we generated five different cloud configuration models, by using a data-center with 64, 128, 256, 384, and 448 physical machines. Figure 14

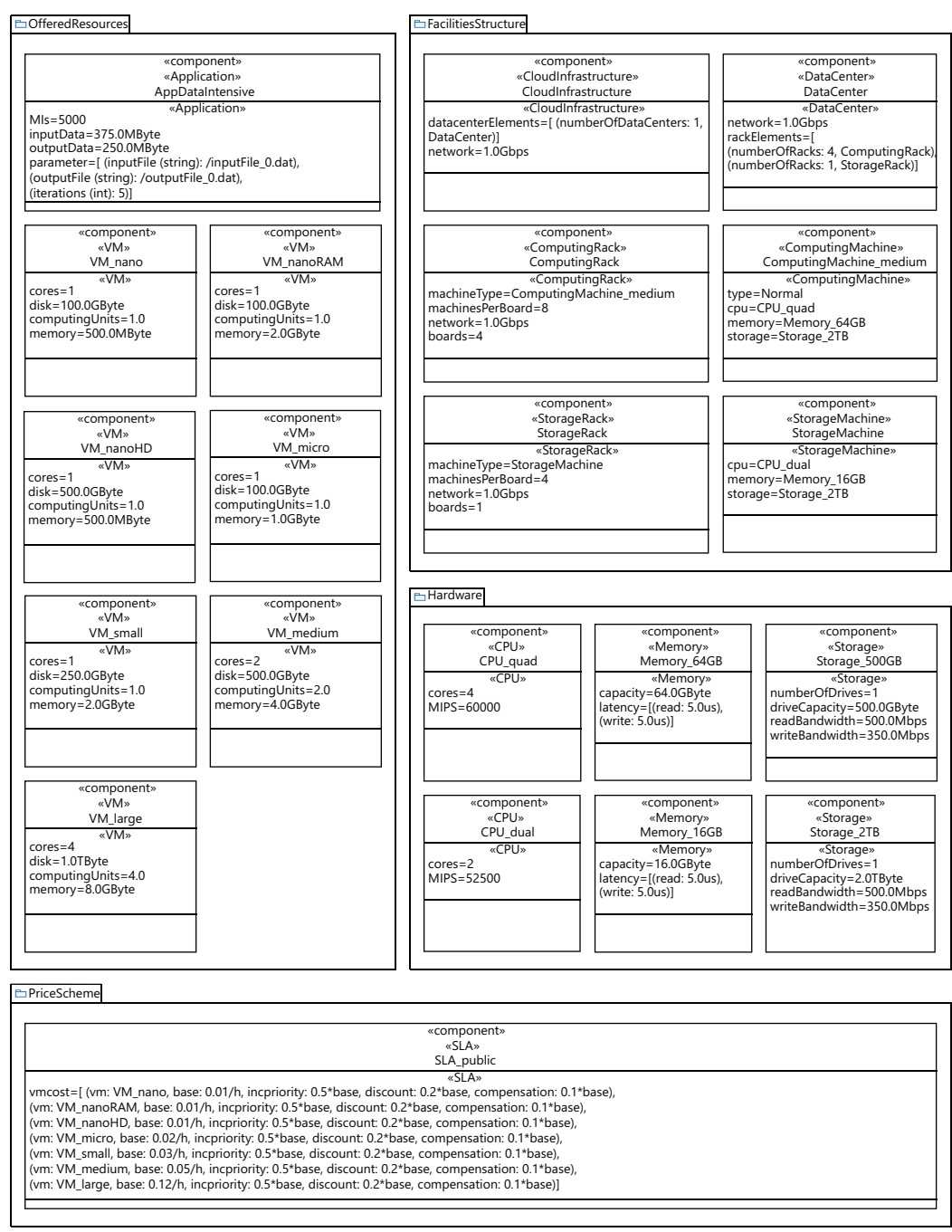

**Figure 14  Profile component diagram defining a cloud infrastructure configuration with 128 nodes.**

shows the configuration of a cloud infrastructure, which is defined by parameterizing the component diagram of the CloudCost profile. This figure in particular shows the configuration of a cloud consisting of one data center with 128 computing machines. Each computing machine in all the clouds modeled has the following characteristics: a 2TB disk, 64GB of RAM, and a quad-core CPU with 60,000 MIPS of computing power. All the components of the data center are interconnected using an Ethernet Gigabit network.

The cloud service provider offers seven different configurations of VMs, and a data-intensive application to be executed on the VMs. We also include the parameters required to model the costs for all the VMs included in the SLA signed by the user. In addition, we designed different synthetic workloads by parameterizing the diagrams of the CloudCost profile. Specifically, we created four different workloads containing 2,000, 5,000, 7,500 and 10,000 users. In these workloads, the percentage of *high-priority* users ranged from 0% to 40% of the total number of users.

Figure 15 shows the sequence diagram, which defines the parameterized behavior of the *regular* user—namely *User_A* in the diagram—when interacting with the cloud service provider. In this case, *User_A* requests a small VM for 2 h and executes an intensive data application on it. The renting time of the initial request is 2 h, but in the event of the user applications not completing their execution in that time, the user can decide to resume the execution for 1 h more. However, it may happen that there are no available VMs with the required features to attend to the initial user request. In such a case, the user can then decide to wait for VMs meeting their requirements to become available, or the user can leave. If the user decides to wait, he subscribes for 24 h, indicating the features required for the VM, with the intention of being notified when a VM with these features becomes available, and then, upon receiving the notification message, the execution of the applications starts. If the subscription time expires and no VM has become available in this period, the user leaves without being able to execute his applications.

## Performance analysis

In this section, we study the income of a cloud service provider with a specific cloud configuration, that is, with a given infrastructure and processing specific workloads. For this purpose, we simulate—using Simcan2Cloud—the execution of four different workloads – containing 2,000, 5,000, 7,500 and 10,000 users—o five different cloud infrastructures, consisting of 64, 128, 256, 384 and 448 physical machines. Furthermore, we consider two possible resource allocation strategies. In the first one, called *NR-first*, *high-priority* users are served first by using non-reserved machines. Only when these resources are not available are the *high-priority* user applications executed on reserved machines. In the second strategy, called *R-first*, *high-priority* users are served first by using the reserved machines. In this case, when there are no more reserved resources available, non-reserved machines are used to attend to their requests.

Figure 16 shows the results obtained from simulating the execution of the workloads on the cloud configuration with 64 computing machines. We indicate the income obtained in relation to the percentage of *high-priority* users (x-axis) and the percentage of reserved machines (y-axis). The income is represented in each square of the chart using the colored scale placed on the right-hand side of the figure. In this particular case, magenta and blue (>400) represent higher incomes, while red and yellow (<0) indicate lower incomes. The graphs on the left show the profits when the *NR-first* strategy is applied, i.e., non-reserved machines are used first to attend to *high-priority* user requests. The graphs on the right show the cloud service provider's incomes when using the *R-first* strategy, i.e., the *high-priority* user requests are served first by using reserved machines.

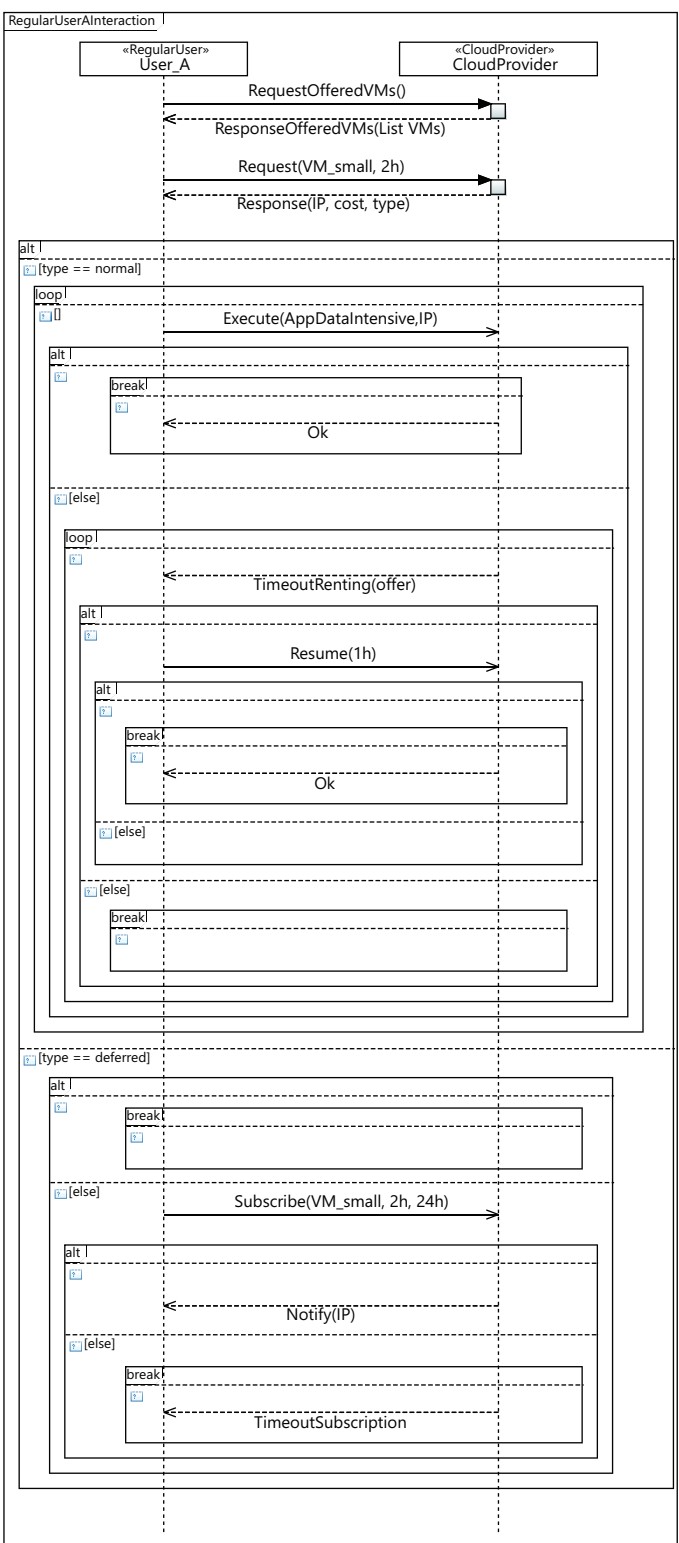

**Figure 15 Profile interaction diagram defining a user workload configuration.**

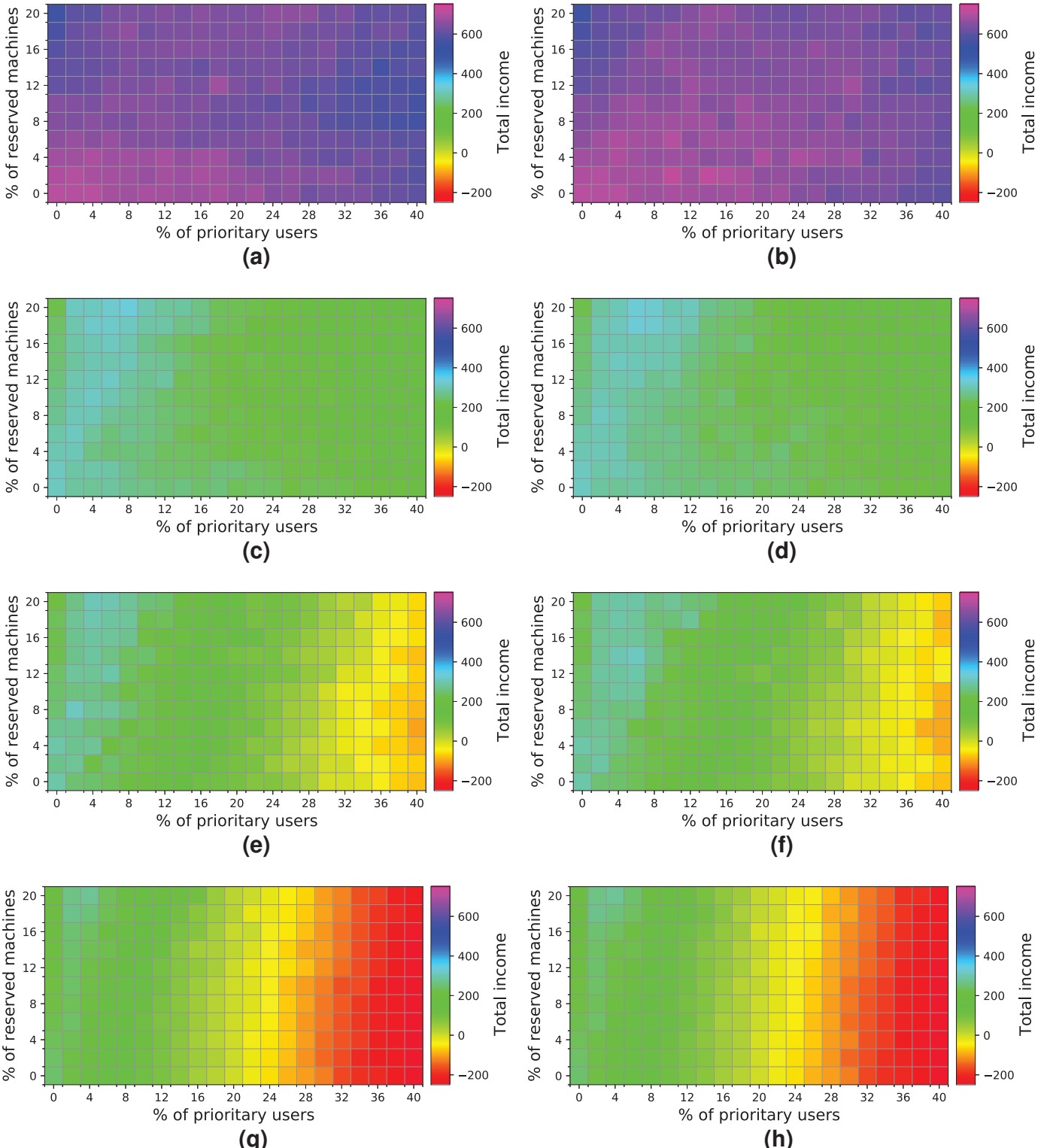

**Figure 16 Overall income of the cloud consisting of 64 machines when processing different workloads.** (A) *NR-first* strategy—2 k users. (B) *R-first* strategy—2 k users. (C) *NR-first* strategy—5 k users. (D) *R-first* strategy—5 k users. (E) *NR-first* strategy—7.5 k users. (F) *R-first* strategy—7.5 k users. (G) *NR-first* strategy—10 k users. (H) *R-first* strategy—10 k users.

Broadly speaking, the results obtained when the two different allocation strategies are used are similar. We can see a slight difference when the cloud processes a workload consisting of 2 k users, in which case the *R-first* strategy (Fig. 16B) provides slightly better results than the *NR-first* strategy (Fig. 16A), as *high-priority* users use the reserved machines from the beginning and then they pay the corresponding surcharge. These graphs also show that the total income decreases as the size of the workload processed increases. This effect is mainly caused by the saturation of the cloud, which is not able to process such a number of users requesting resources and, consequently, the number of unattended *high-priority* users increases significantly, which in turn increases the number of compensations. In this particular case, the size of the cloud clearly limits the overall income, this therefore being the main bottleneck of the system. The best-case scenario using this cloud configuration reaches an income of—approximately—600 monetary units when the workload of 2k users is processed.

Next, we analyze how a cloud consisting of 128 machines processes the workloads (Fig. 17). We observe some similarities with the previous experiment. First, both allocation strategies obtain almost the same results. Second, the highest income is obtained when a workload consisting of 2 k users is processed (Figs. 17A and 17B). Third, the income decreases when the size of the workload and the percentage of *high-priority* users increases—especially with a high number of reserved machines – because there are a large number of users leaving the system without being served. In contrast to the previous cloud providing only 64 machines, this cloud obtains higher incomes, which is mainly thanks to increasing the number of physical machines. In this case, there is no negative income. However, the cloud is still saturated and, therefore, the number of reserved machines is not enough to allow these users to be attended to, so compensations reduce the final incomes. The best-case scenario generates an income of – approximately—1,600 units when the workload of 2 k users is processed.

Figure 18 shows the results obtained for a cloud with an infrastructure containing 256 physical nodes. In this case, processing workloads containing requests from 5 k, 7.5 k and 10 k users obtains better results than those when the workload of 2 k users is processed. Hence, increasing the physical resources has a significant impact on the overall income.

It is important to note that these charts clearly show a turning point in the overall income when the workload containing 7.5 k users is processed. Note that with the *R-first* strategy, high-priority users are served first using the reserved machines. Thus, *regular* users have more non-reserved machines available, as long as the system is not saturated, so they do not have to compete with *high-priority* users in this case. We can also see that from 7.5 k users upwards (Figs. 18E, 18F, 18G and 18H) the cloud becomes saturated again with a high number of *high-priority* users. This case provides the best results for this cloud (Figs. 18E and 18F). However, in this particular case, the two allocation strategies lead to different results. Figure 18E shows that as the number of *high-priority* users increases (using the *NR-first* strategy), the income is only maintained with a percentage of reserved nodes lower than 14%. However, when the *R-first* strategy is used (Fig. 18F) we can see that the income can be maintained with a percentage of reserved machines greater than

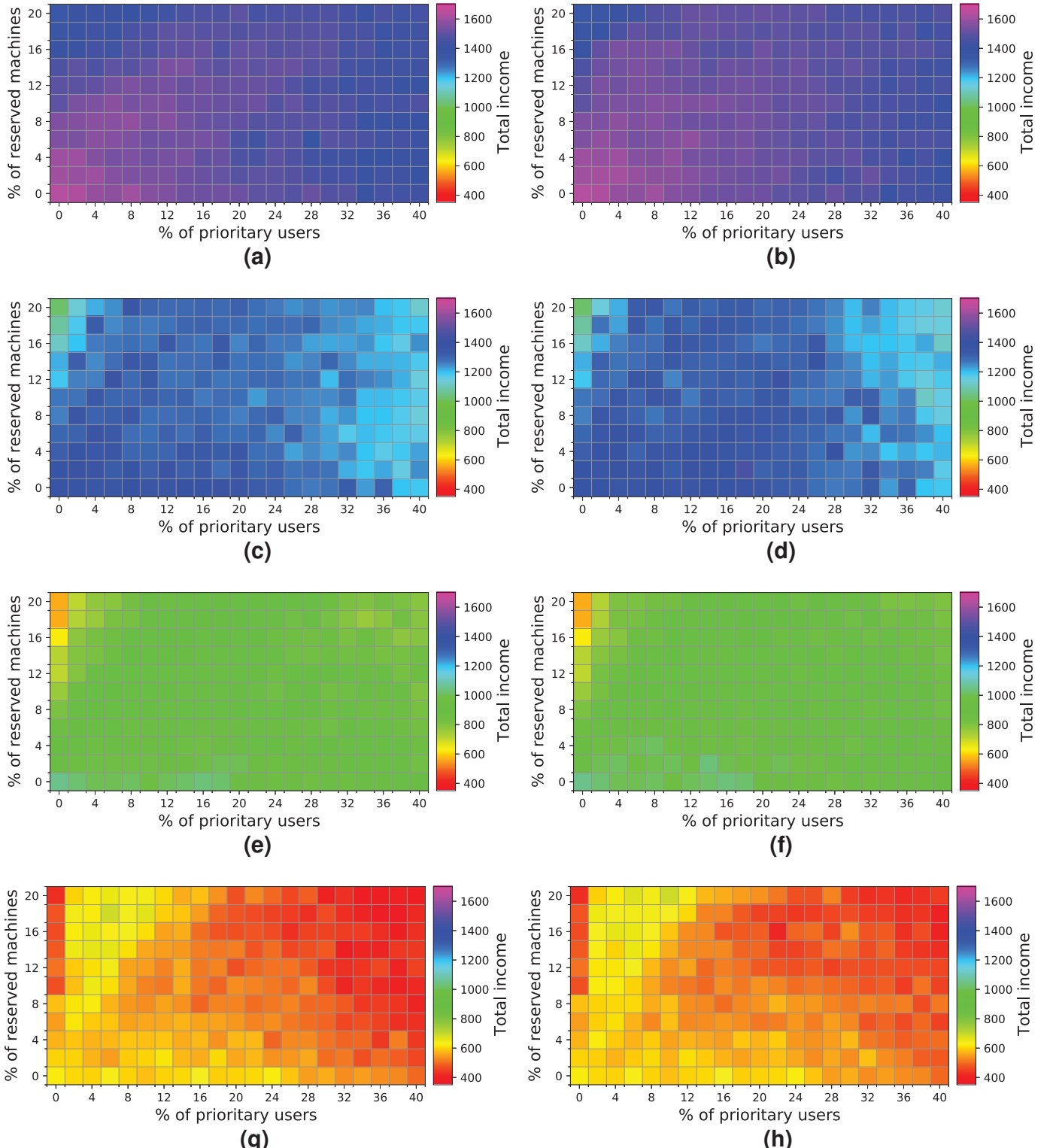

**Figure 17 Overall income of the cloud consisting of 128 machines when processing different workloads.** (A) *NR-first* strategy—2 k users. (B) *R-first* strategy—2 k users. (C) *NR-first* strategy—5 k users. (D) *R-first* strategy—5 k users. (E) *NR-first* strategy—7.5 k users. (F) *R-first* strategy—7.5 k users. (G) *NR-first* strategy—10 k users. (H) *R-first* strategy—10 k users.

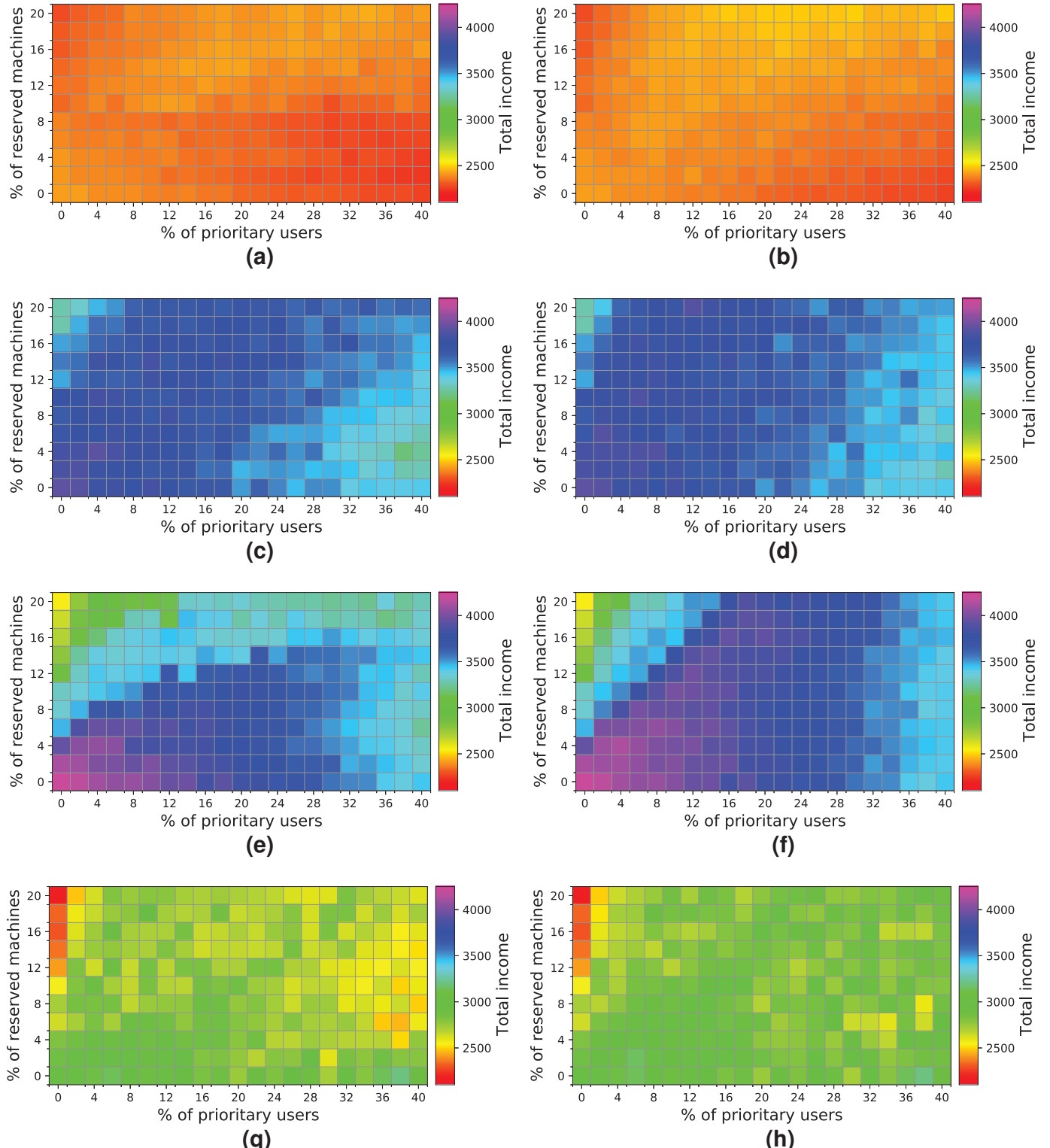

**Figure 18 Overall income of the cloud consisting of 256 machines when processing different workloads.** (A) *NR-first* strategy—2 k users. (B) *R-first* strategy—2 k users. (C) *NR-first* strategy—5 k users. (D) *R-first* strategy—5 k users. (E) *NR-first* strategy—7.5 k users. (F) *R-first* strategy—7.5 k users. (G) *NR-first* strategy—10 k users. (H) *R-first* strategy—10 k users.

14% (see purple area), so the global income—in this particular case—is better with the *R-first* strategy.

The best-case scenario in this cloud is obtained when the *R-first* strategy is used to process a workload consisting of 7.5 k users, and it generates an income of 4,000 monetary units. Note that the saturation of the cloud reduces the income when the number of users increases (Figs. 18G and 18H).

Figure 19 shows the results obtained for a cloud configuration with 384 physical nodes. Processing the workloads of 2 k and 5 k users obtains similar results to those for the previous clouds, that is, the *R-first* strategy provides slightly better results—since *high-priority* users pay a surcharge for using the reserved machines—than the *NR-first* strategy. In these cases, the cloud service provider income is very similar regardless of the percentage of *high-priority* users. This situation occurs because the workload can be processed without the cloud becoming saturated, and thus most users are served using only non-reserved resources.

These charts show a turning point between 7.5 k and 10 k users. When the percentage of *high-priority* users is low, as the percentage of reserved machines increases, the profits decrease due to the *regular* users that have to leave the system (see the light blue squares in the upper-left corner). In the same way, when the percentage of *high-priority* users is high and the percentage of reserved machines is low, the profits also decrease due to the compensations. These effects can be observed in Figs. 19E, 19F, 19G and 19H.

In this cloud, in contrast to the previous ones, the highest incomes (>6,000 monetary units) is obtained when processing a workload consisting of 10 k users and using the *R-first* strategy.

The last experiment (Fig. 20) shows the results obtained for a cloud configuration with 448 physical nodes. In this case, we observe the same tendency as in the previous experiment (Fig. 19), that is, the two allocation policies obtain the same results when the cloud is not saturated (Figs. 20A, 20B, 20C and 20D), and the *R-first* strategy provides slightly better results when the cloud reaches the saturation point (Figs. 20E, 20F, 20G and 20H). In this case, however, increasing the number of physical machines generates an improvement in the overall income, reaching 8,000 monetary units when the cloud processes a workload of 10 k users (Fig. 20H).

## Discussion of the results

In this section, we provide a detailed discussion of the results obtained in the experimental study and draw some interesting conclusions.

After a careful analysis of the results, we discover that the size of the cloud, that is, the number of physical machines, has a significant impact on the overall income, and therefore it should be dimensioned in proportion to the workload to be processed. The CloudCost profile allows us to model the behavior of the users and, then, to simulate the behavior of cloud systems when processing different workloads, so as to determine the turning point at which the cloud increases the overall income. It is therefore desirable that the percentage of reserved machines (y-axis) is balanced in proportion to the percentage of *high-priority* users (x-axis) requesting resources from the cloud.

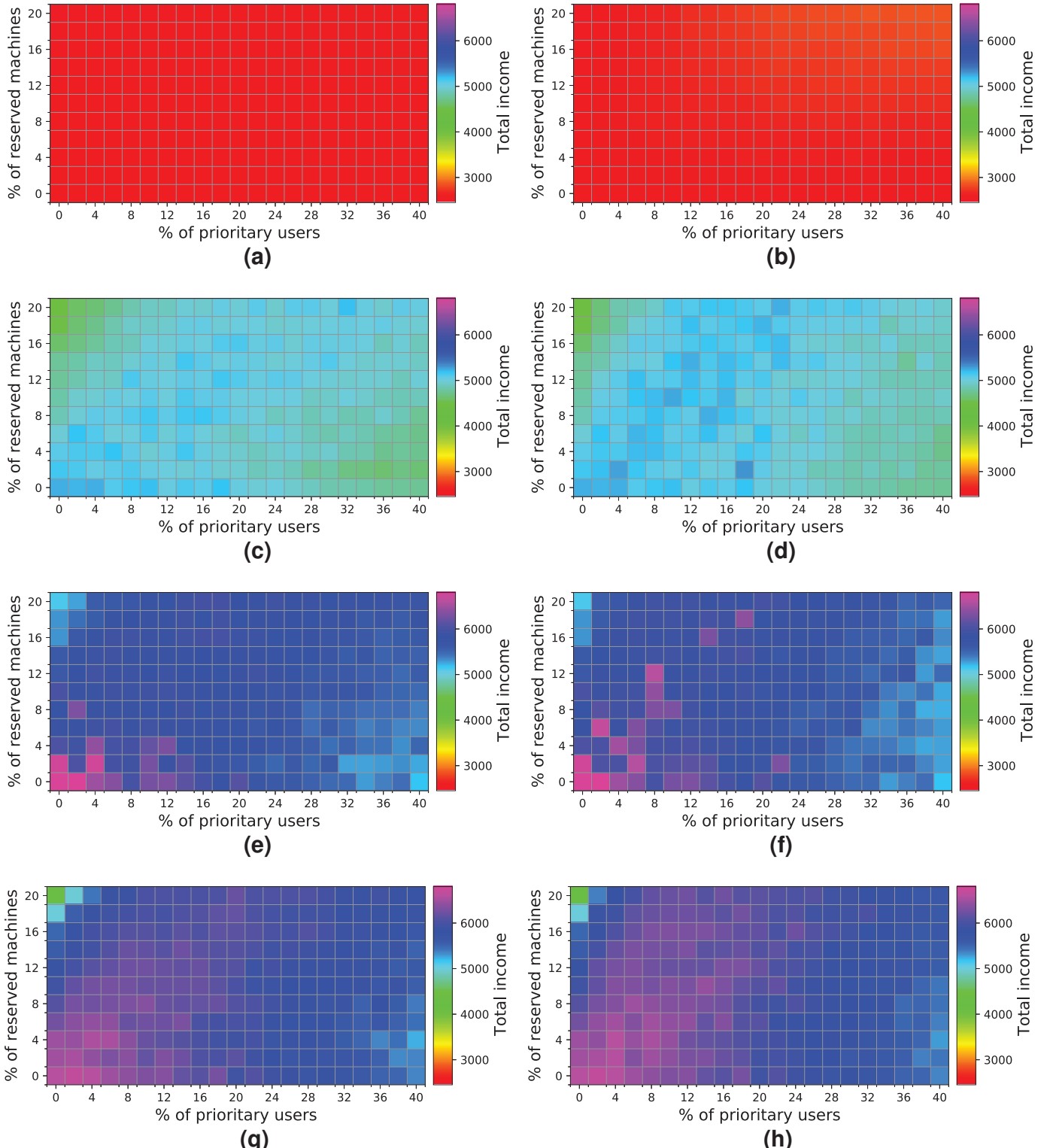

**Figure 19 Overall income of the cloud consisting of 384 machines when processing different workloads.** (A) *NR-first* strategy—2 k users. (B) *R-first* strategy—2 k users. (C) *NR-first* strategy—5 k users. (D) *R-first* strategy—5 k users. (E) *NR-first* strategy—7.5 k users. (F) *R-first* strategy—7.5 k users. (G) *NR-first* strategy—10 k users. (H) *R-first* strategy—10 k users.

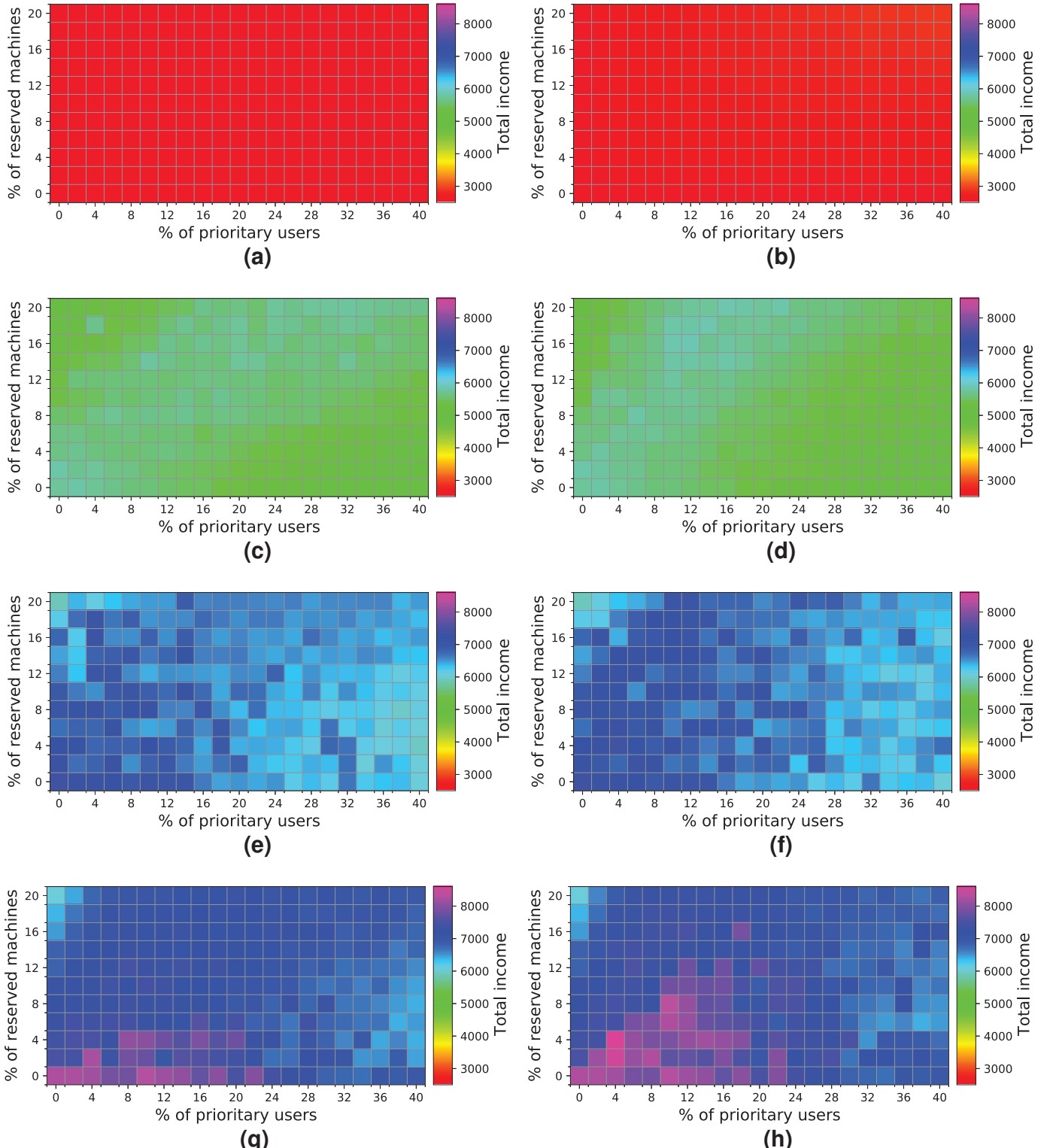

**Figure 20 Overall income of the cloud consisting of 448 machines when processing different workloads.** (A) *NR-first* strategy—2 k users. (B) *R-first* strategy—2 k users. (C) *NR-first* strategy—5 k users. (D) *R-first* strategy—5 k users. (E) *NR-first* strategy—7.5 k users. (F) *R-first* strategy—7.5 k users. (G) *NR-first* strategy—10 k users. (H) *R-first* strategy—10 k users.

Note from the experiments that when the cloud is not saturated (all the users' requests are served), the two allocation policies obtain similar results. However, when the cloud is saturated, the *R-first* strategy provides slightly better results than the *NR-first* strategy. The experiments clearly indicate the saturation point of the cloud when increasing the number of users requesting resources from the cloud. This is an important aspect that must be carefully analyzed by the cloud service provider in order to adapt the size of the cloud to the load that is to be processed.

In addition, we have discovered a boundary in the percentage of *high-priority* users that clearly limits the overall income. In these experiments, when the percentage of *high-priority* users increases by 20%, the income decreases. The results show that increasing the number of *high-priority* users could potentially harm the cloud service provider's profit when the resources are not appropriately assigned. However, the cloud service provider could alleviate this situation by reserving machines to provide resources exclusively to *high-priority* users.

We can conclude that, in most scenarios, having a good ratio of reserved machines to attend to *high-priority* users is key to increasing the cloud provider's overall income.

Finally, as future work, we intend to perform a mediation effect analysis (*Baron & Kenny, 1986*; *Shrout & Bolger, 2002*) on the simulation results obtained. The regression sets will be the number of *high-priority* users requesting resources from the cloud (X), the number of reserved nodes for *high-priority* users (M), and the cloud service provider's profit (Y). Thus, the goal of this study will be to analyze the impact of *M* on the causal effect that *X* has on *Y*, i.e. to conclude what the effect is of varying the number of nodes reserved for *high-priority* users on the effect of *X* on *Y*.

Note: The data and the results obtained for the cloud configurations considered can be found in Supplemental Data S1.

## CONCLUSIONS AND FUTURE WORK

In this paper, we have proposed the CloudCost UML profile as a means for modeling a cloud infrastructure and the user interactions with the cloud service provider, with two different SLAs for the users (*regular* and *high-priority*). *Regular* users can wait for the resources they need and subscribe to the cloud provider to be informed when the resources become available. In contrast, *high-priority* users need the resources immediately, and some physical resources are then reserved in order to be able to attend to their requests. The CloudCost profile allows the modeling of complex cloud scenarios, in which we can represent the underlying cloud infrastructure, the cost of the resources, and the workload submitted by the users, taking into account the SLA they have signed. A complete case study involving the modeling and evaluation of different cloud scenarios has been presented to examine the impact of the parameters considered. From this case study, we have concluded that it is beneficial to reserve some machines to serve the *high-priority* user requests. Another conclusion is that better results are obtained with the strategy that first assigns the reserved machines to *high-priority* users. This strategy actually has a positive impact on the cloud service provider's income in some scenarios. Furthermore,

this resource allocation strategy did not produce negative consequences for the cloud service provider's profits in any scenario.

For future work, we plan several lines of research. We intend to enrich the profile by including other possible SLAs, studying, for instance, the procurement schemes of Amazon Web Services (Amazon AWS (*Amazon, 2021*)), as well as combinations of them. We also plan to extend the spectrum of possible cloud configurations, not only using a different number of physical machines but also broadening the range of configurations for the hardware such as, among others, the CPUs and the disk space of the hosts. Thus, we expect to obtain relevant and useful conclusions from these new studies.

### Funding
This work was supported by the Spanish Ministry of Science and Innovation (co-financed by European Union FEDER funds) project "FAME (Formal modeling and advanced testing methods. Applications to medicine and computing systems) and MASSIVE (Engineering adaptive software by and for the people in a highly connected world)", references RTI2018-093608-B-C32 and RTI2018-095255-B-I00. There was also support from the Junta de Comunidades de Castilla-La Mancha project SBPLY/17/180501/000276/01 (cofunded with FEDER funds, EU), the Region of Madrid (grant number FORTE-CM, S2018/TCS-4314), and the Madrid Government (Comunidad de Madrid-Spain) under the Multiannual Agreement with the Complutense University as part of the Program to Stimulate Research for Young Doctors in the context of the V PRICIT (Regional Programme of Research and Technological Innovation) under grant PR65/19-22452. The funders had no role in study design, data collection and analysis, decision to publish, or preparation of the manuscript.

### Grant Disclosures
The following grant information was disclosed by the authors:
Spanish Ministry of Science and Innovation.
"FAME": RTI2018-093608-B-C32.
"MASSIVE": RTI2018-095255-B-I00.
Junta de Comunidades de Castilla-La Mancha: SBPLY/17/180501/000276/01.
Region of Madrid: FORTE-CM and S2018/TCS-4314.
Madrid Government (Comunidad de Madrid-Spain) under the Multiannual Agreement with the Complutense University as part of the Program to Stimulate Research for Young Doctors in the context of the V PRICIT (Regional Programme of Research and Technological Innovation) under grant PR65/19-22452.

### Competing Interests
M. Emilia Cambronero is an Academic Editor for PeerJ.

## Author Contributions

- M. Emilia Cambronero conceived and designed the experiments, analyzed the data, performed the computation work, prepared figures and/or tables, authored or reviewed drafts of the paper, and approved the final draft.
- Adrián Bernal conceived and designed the experiments, performed the experiments, analyzed the data, performed the computation work, prepared figures and/or tables, authored or reviewed drafts of the paper, and approved the final draft.
- Valentín Valero conceived and designed the experiments, analyzed the data, performed the computation work, prepared figures and/or tables, authored or reviewed drafts of the paper, and approved the final draft.
- Pablo C Cañizares conceived and designed the experiments, analyzed the data, performed the computation work, prepared figures and/or tables, authored or reviewed drafts of the paper, and approved the final draft.
- Alberto Núñez conceived and designed the experiments, analyzed the data, performed the computation work, prepared figures and/or tables, authored or reviewed drafts of the paper, and approved the final draft.

## Data Availability

The data obtained from the simulations and source code are available in the Supplemental Files.

## Supplemental Information

Supplemental information for this article can be found online at http://dx.doi.org/10.7717/peerj-cs.513#supplemental-information.

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
