# Peer review of "Profiling SLAs for cloud system infrastructures and user interactions"

_PeerJ Computer Science, doi:10.7717/peerj-cs.513_

## Round 0.1 · original submission · Major Revisions

Dear Authors,

As per the received reviews, I have to make the decision as major revisions. I hope you will fully address all the reviews before re-submission. Good luck.

Reviewer 1 ·

Basic reporting

The writing and structure must be improved

Experimental design

Introduction seems too long , it must be shortened and concised with relevant studies only.
It is very hard to find the novelty or any scientific contribution in the paper, please justify it and provide a separate section.

Validity of the findings

Which testbed you have used for cloud configuration?
It would be nice if you can provide validation and state of the art comparison
Please provide the link of your tool
Cloud configuration performance and analysis is more theoretical and boring
It is suggested to add more case studies to justify the study
There are many cited online references, I suggest to reduce the citation of online references, if unavoidable then please give access time/data with them. E.g. Last access on 07-December-2020.
Figures 13 , 14 are not readable, please provide clear figures.

Additional comments

please revise carefully according to my comments above

Reviewer 2 ·

Basic reporting

This study attempts to investigate the role of SLAs on IaaS. In addition, this study also attempts to understand the benefits afforded to SLA and clouds, provide conceptual insights into it, and test the proposed model. In general, the paper was even well written and interesting, but the logic does not flow well and needs to supplement or enrich.

Experimental design

The results part needs to provide more information to audiences, not just pronounce the significant coefficients in tables. Besides, the simulation results need to show in some other figures, and the causal relationship need to explain systematically and under your inferences and hypotheses. Authors should explain it with a standalone paragraph and tell us how to test the mediation effect.

Validity of the findings

It seems valid.

Additional comments

* With respect to writing style, the entire paper strongly requires to be reviewed by a native English speaker. There many grammatical issues, such as missing articles, missing verbs, and mixed-up singulars/plurals.
* The motivation of the work is not clearly presented especially in the abstract. They should better motivate their work so that they could be able to position the work to the concurrent solutions.
*There are some style and spelling mistakes. Some sentences are repeated with almost the same words.
* Font sizes need to be consistent.
* There is no clear indication of what the model is.
* The implications of the study should be rearranged according to the implication of theory, methods, and practice. Thus, it would be easy for the readers to see the volume of the contribution of the paper. It should explain how this study departs from previous research particularly in terms of new knowledge and theory.
*The explanation of the data set is superficial and needs to put more explanation.
* The methodology is simplistic and poorly documented.
* The Abstract is absolutely not informative.
* The literature is not critically written and more towards a positive tone. Contradiction stands need to be synthesized as well. Or else the literature review is just merely reporting, It needs to be more analytical, decisive, and avoid a positive tone all the way. Tabulate the literature to enable the readers to see the volume of works done in this area to date.
* Also the motivational approach of cloud and SLA would be important to integrate shortly into the introduction and related works. Just a few studies to mention:
Cloud service composition using an inverted ant colony optimisation algorithm. 13(4), 257-268.
A new genus of oryzomyine rodents (Cricetidae, Sigmodontinae) with three new species from montane cloud forests, western Andean cordillera of Colombia and Ecuador. 8, e10247.
Human resources ranking in a cloud-based knowledge sharing framework using the quality control criteria.
Epiphytic bryophyte biomass estimation on tree trunks and upscaling in tropical montane cloud forests. 8, e9351.
SLA-aware and energy-efficient VM consolidation in cloud data centers using robust linear regression prediction model. 7, 9490-9500.
A new agent-based method for QoS-aware cloud service composition using particle swarm optimization algorithm. 10(5), 1851-1864.
SCelVis: exploratory single cell data analysis on the desktop and in the cloud. 8, e8607.
Collaborative SLA and reputation-based trust management in cloud federations. 100, 498-512.
A cloud service composition method using a trust‐based clustering algorithm and honeybee mating optimization algorithm. 33(5), e4259.
Enforcing trustworthy cloud SLA with witnesses: A game theory–based model using smart contracts. e5511.

---

## Round 0.2 · accepted · Accept

Congratulations, the reviewers are satisfied with the revised version of the manuscript and recommended for publication.

Reviewer 1 ·

Basic reporting

no comments

Experimental design

no comments

Validity of the findings

no comments

Additional comments

Good luck